



**Niche differentiation of ammonia and nitrite oxidizers along a salinity gradient**
**from the Pearl River estuary to the South China Sea**
Lei Hou[1,2,†], Xiabing Xie[1,†], Xianhui Wan[1], Shuh-Ji Kao[1,2], Nianzhi Jiao[1,2], Yao Zhang[1,2]
[1]State Key Laboratory of Marine Environmental Science, Xiamen University, Xiamen 361101, China
[2]College of Ocean and Earth Sciences, Xiamen University, Xiamen 361101, China
*Correspondence to*: Yao Zhang (yaozhang@xmu.edu.cn)
[†]Contributed equally

# Abstract

The niche differentiation between ammonia and nitrite oxidizers are controversial because they display disparate patterns in estuarine, coastal, and oceanic regimes. We analyzed ammonia-oxidizing archaea (AOA) and $\beta$-proteobacteria (AOB) *amo*A genes, nitrite-oxidizing bacteria (NOB) *nxr*B and 16S rRNA genes, and nitrification rates to identify their niche differentiation along a salinity gradient from the Pearl River estuary to the South China Sea. The archaeal *amo*A genes were generally more abundant than the $\beta$-AOB *amo*A genes; however, AOB more clearly attached to particles compared with AOA in the upper reaches of the Pearl River estuary. The NOB *Nitrospira* had higher abundances in the upper and middle reaches of the Pearl River estuary, while *Nitrospina* was dominant in the lower estuary. In addition, AOB and *Nitrospira* could be more active than AOA and *Nitrospina* since significantly positive correlations were observed between their gene abundance and the nitrification rate in the Pearl River estuary. There is a coupling of ammonia and nitrite oxidizers in the hypoxic waters of the estuary, suggesting metabolic interactions between them. Phylogenetic analysis further revealed that the AOA and NOB *Nitrospina* subgroups can be separated into different niches based on their adaptations to substrate levels. Water mass mixing is apparently crucial in regulating the distribution of nitrifiers from the estuary to open ocean. However, when eliminating water mass effect, the substrate availability and the nitrifiers' adaptations to substrate availability via their ecological strategies essentially determine their niche differentiation.



## 1 Introduction

Nitrification, the biological oxidation of ammonia to nitrate (the largest pool of fixed inorganic nitrogen
in water bodies), is a fundamental process in the nitrogen cycle and plays a key role in estuarine and
marine ecosystems. Nitrification includes both ammonia and nitrite oxidation, which are catalyzed by
different microorganisms who may occupy broad niches in estuarine and marine environments. The first
nitrification step, ammonia oxidation, is predominantly carried out by ammonia-oxidizing archaea
(AOA) belonging to the phylum *Thaumarchaeota*, and ammonia-oxidizing bacteria (AOB).
*Thaumarchaeota* are more adapted to ammonia-limited oligotrophic conditions than AOB (Erguder et
al., 2009; Martens-Habbena et al., 2009). The gene coding for ammonia monooxygenase subunit A
(*amo*A) has been widely applied as a functional marker gene for ammonia oxidizers (Juretschko et al.,
1998; Francis et al., 2005; Leininger et al., 2006; Tourna et al., 2008; Gubry-Rangin et al., 2011; Pester
et al., 2012).

14        In sharp contrast to ammonia oxidation, nitrite oxidation, which is the second step in nitrification,

has been investigated less in estuarine and marine ecosystems, despite bacterial nitrite oxidation being
the only biochemical reaction known to form nitrate in aerobic conditions. In addition, a considerable
fraction of recycled nitrogen or reduced nitrate is re-oxidized back to nitrate via nitrite oxidation in
oxygen minimum zones (OMZs; Füssel et al., 2012; Beman et al., 2013; Casciotti et al., 2013; Bristow
et al., 2016). Nitrite oxidation is catalyzed by nitrite-oxidizing bacteria (NOB). To date, seven genera of
NOB have been described: *Nitrospira*, *Nitrospina*, *Nitrococcus*, *Nitrobacter*, *Nitrolancea*, *Nitrotoga*,
and *Candidatus* Nitromaritima (Spieck and Bock 2005; Alawi et al., 2007; Sorokin et al., 2012; Ngugi
et al., 2016). Members of the genus *Nitrospira* appear to be the most diverse and widespread in a
diverse range of habitats (Daims et al., 2001; Lücker et al., 2010), while *Nitrospina* are reported to be
restricted to marine environments (Lücker et al., 2013). *Nitrobacter* and *Nitrococcus* are less abundant
and confined mainly to freshwater/estuarine and oceanic settings, respectively (Koops and
Pommerening-Roser, 2001; Füssel et al., 2012). *Nitrotoga* has been detected in a marine recirculation
aquaculture system (Keuter et al., 2017). *Candidatus* Nitromaritima were recently identified based on
metagenomic data in Red Sea brines (Ngugi et al., 2016). The gene encoding subunit beta of nitrite
oxidoreductase (*nxr*B) is a functional and phylogenetic marker for NOB (Wertz et al., 2008; Pester et al.,
2013; Schwarz, 2013). However, there is a *nxr*B-targeting primer sets coverage limitation, so that the
NOB 16S rRNA gene has been used as a useful marker for quantifying the NOB community in various
ecosystems (Mincer et al., 2007; Nunoura et al., 2015).
The niche differentiation between ammonia and nitrite oxidizers is controversial because it
displays disparate patterns and partnerships in estuarine, coastal, and oceanic regimes. A gradient from
an estuary to the ocean, with distinct distribution patterns of various nutrient species, may provide
diverse niches for the coexistence of microbial species (Martens-Habbena et al., 2009). It is thus an



ideal system to study the niche differentiation of AOA, AOB and NOB and major controlling factors.

2       The Pearl River is the largest river in southern China. Human activity has seriously affected the

regional environment over the past few decades. A persistent oxygen depletion zone was found in the
upper reaches of the Pearl River estuary (PRE) (He et al., 2014), which has been attributed to organic
matter degradation and nitrification (Dai et al., 2006; 2008; He et al., 2010). The Pearl River drains into
the northern part of the tropical oligotrophic South China Sea (SCS), the largest deep (maximum water
depth of ~5560 m) semi-enclosed marginal sea in the western Pacific Ocean. Thus, the northern SCS is
influenced by large amounts of freshwater and nutrient input from the Pearl River. The Southeast Asia
Time-Series Study (SEATS) site, the only active time-series station located in a marginal sea (Wong et
al., 2007; Zhang et al., 2014b), is situated in the SCS central basin (18°N, 116°E) at a depth of 3850 m
and characterized by low nutrient levels. This environment, spanning the PRE to the SCS, provides a
great opportunity to explore the microbial groups driving ammonia and nitrite oxidation within
complicated biogeochemical settings.
In this study, the diversity of AOA and AOB *amo*A and NOB *nxr*B genes was investigated by clone
libraries, and distributions of AOA and AOB *amo*A and NOB 16S rRNA genes were quantified by
quantitative polymerase chain reaction (qPCR) along a salinity gradient from the PRE to the SCS (Fig.
1). Moreover, nitrification rates were determined in the PRE using $^{15}$N-labeled ammonium (Sigman et
al., 2001). The objectives of this study were to (1) investigate the spatial patterns of diversity and



abundance of AOA, AOB, and NOB, (2) explore the niche differentiation and relationship between
AOA, AOB, and NOB, and (3) explain the possible environmental parameters governing niche
differentiation.

## 2 Materials and methods

### 2.1 Study sites and sampling

Twelve sites (P1–P12) along the PRE as well as the SEATS station in the SCS central basin were
sampled during two summer research cruises in July–August 2012 and September 2014 (Fig. 1). Both
the surface (1 m) and bottom waters (1.5–3.5 m above the seafloor) were sampled at the 12 PRE sites
(Table S1); there were exceptions for sites P2, P3 and P4 where only the bottom water was sampled and
P6 where only the surface water was sampled. The SEATS site was sampled at 75 m, 200 m, 800 m, and
3000 m water depth. Water samples were collected using a conductivity, temperature, and depth (CTD)
rosette sampling system fitted with Go-Flo bottles (SBE 9/17 Plus; SeaBird Inc, USA). A total of 44
samples were subjected to gene analysis. A total of 10 samples from the bottom waters of sites P2–10
and the surface water of site P9 were amended with $^{15}$N-labeled ammonium to measure nitrification
rates.

### 2.2 Biogeochemical parameters

Temperature, salinity, and depth data were obtained from the CTD system. Dissolved oxygen (DO)
concentrations were directly measured onboard via the Winkler method (Carpenter, 1965). Water
samples for inorganic nutrients such as nitrate, nitrite, phosphate, and silicate were filtered through 0.45
μm cellulose acetate membranes and then analyzed onboard. Ammonium was analyzed by the
indophenol blue spectrophotometric method (Pai et al., 2001). Nitrite and nitrate were measured with a
four-channel continuous flow Technicon AA3 Auto-Analyzer (Bran-Lube GmbH, Germany) (Han et al.,
2012). Water samples for total suspended material (TSM) were filtered on to pre-combusted and
pre-weighed glass fiber filter membranes (Whatman), and then stored at -20°C until weighing in the
laboratory.
**2.3 DNA extraction**
One liter of water from each PRE sample was filtered through 3 μm and then 0.22 μm pore-size
polycarbonate membranes (47 mm diameter; Millipore) at a pressure of <0.03 MPa to retain the
particle-associated (PA) communities (size fraction >3 μm) and free-living (FL) communities (size
fraction 0.22–3 μm) for DNA extraction. For the SCS samples, 2 or 4 liter water samples were directly
filtered through 0.22 μm pore-size polycarbonate membranes (47 mm diameter; Millipore) for DNA
extraction. All of the polycarbonate membranes were flash frozen in liquid nitrogen and then stored at
-80°C until further analysis. DNA was extracted using the UltraClean Soil DNA kit (MoBio, San Diego,



CA, USA) following the manufacturer's protocols. Concentration and purity of the genomic DNA were
checked with a NanoDrop spectrophotometer (Thermo Scientific 2000/2000c) (Johnson, 1994).
**2.4 PCR, cloning, sequencing, and phylogenetic analysis**
Archaeal and *β*-proteobacterial *amo*A genes were amplified using primer sets Arch-amoAF and
Arch-amoAR (Francis et al., 2005), and amoA-34F and amoA-2R (Kim et al., 2008), respectively.
*Nitrospira*, *Nitrospina*, *Nitrobacter*, and *Nitrococcus nxr*B genes were amplified. Primer set sequences,
PCR reaction mixtures and conditions for each functional gene are listed in Table S2. We designed
primers for the *Nitrospina nxr*B gene based on two *nxr*B gene sequences of *Nitrospina gracilis* 3/211
using PREMIER software (Biosoft International, USA). Forward primer nxrBNF (5'-GGG CGA CCA
GAT GGA AAC-3') and reverse primer nxrBNR (5'-GGG CCG GAC ATA GAA AGG-3') target the
771–788 and 1237–1254 nucleotide regions, respectively, of the *nxr*B gene in *N. gracilis* 3/211. The
specificity of this designed primer pair was tested by BLASTn searches in the GenBank database. The
amplified target fragments were purified using an agarose gel DNA purification kit (Takara, Dalian,
China), ligated into the pMD18-T vector (Takara), and transformed into competent cells of *Escherichia*
*coli* DH5α. Positive clones were randomly selected for sequencing using an ABI model 3730 automated
DNA sequence analyzer with BigDye terminator chemistry (Perkin-Elmer, Applied Biosystems, USA).

18        All gene sequences were grouped into operational taxonomic units (OTUs) based on a 5%


sequence divergence cutoff (Wankel et al., 2011; Pester et al., 2013; Rani et al., 2017) by using the
DOTUR program (Schloss and Handelsman, 2005). Representative nucleotide sequences were analyzed
with the BLASTn tool to get the closest reference sequences. Neighbor-joining phylogenetic trees were
constructed with MEGA 5 software using a Maximum Composite Likelihood model for archaeal *amo*A
gene sequences (Zhang et al., 2014a) and Jukes-Cantor model for *Nitrospira* and *Nitrospina nxr*B gene
sequences (Pester et al., 2013). A phylogenetic tree was not constructed for bacterial *amo*A gene and
*Nitrobacter nxr*B gene sequences because too few sequences were retrieved. The *Nitrococcus nxr*B gene
was not amplified successfully from these samples.
**2.5 Quantitative PCR amplification**
Abundances of the archaeal and *β*-proteobacterial *amo*A genes, and *Nitrospira* and *Nitrospina* 16S
rRNA genes were quantified using a qPCR method and a CFX 96™ (BIO-RAD, Singapore) real-time
system. Standard curves were constructed for archaeal and *β*-proteobacterial *amo*A genes using plasmid
DNA from clone libraries. For *Nitrospira* and *Nitrospina* 16S rRNA genes, the target DNA fragments of
the pure cultured strains were used. Quantitative PCR reactions were performed in triplicate and
analyzed against a range of standards (1 to $10^7$ copies per μl). Primer pair sequences, qPCR mixtures
and conditions for each gene are listed in Table S3. The efficiencies of qPCR amplification ranged from
90% to 104% with $R^2$ >0.99. The specificity of the qPCR reactions was checked by melting curve
analysis and agarose gel electrophoresis. The uncertain products were sequenced to confirm their
veracity. Inhibition tests were performed by 2-fold and 5-fold dilutions of all samples and we concluded
that our samples were not inhibited.
**2.6 $^{15}$N-labeled nitrification rate measurements**
Nitrification rates (oxidation of ammonia to nitrate) were measured using the stable isotope tracer
method described in Hsiao et al. (2014) with minor modifications. Briefly, six 115 mL narrow-necked
gas-tight glass bottles were overflowed to more than twice their volume with seawater and sealed
without headspace. Then, a syringe was used to replace 1 mL of sample with the $^{15}$N-NH$_4^+$ tracer (98%
of $^{15}$N atoms, Sigma-Aldrich) to attain a final tracer concentration of 1 μmol L$^{-1}$, which accounted for
1%–10% of total ammonia concentration in the upper PRE (P2–6, *in situ* rates of nitrification can be
estimated) and >10% in the middle and lower reaches (P7–10, potential nitrification rates were
obtained). Three bottles were filtered immediately after the tracer injection through 0.22 μm
polycarbonate filters to represent the initial conditions. The remaining three bottles were kept in the
dark for 6 h under *in situ* temperature (±1°C) using a temperature control incubator. The incubations
were terminated by filtering through 0.22 μm polycarbonate membranes, and the filtrate was frozen at
-20°C until laboratory analysis.
Ammonium, nitrite, and nitrate were detected as described above. The detection limits for



ammonium, nitrite and nitrate were 0.16, 0.03 and 0.05 $\mu$mol L$^{-1}$, respectively. The $\delta^{15}$N of NO$_X^-$ (NO$_2^-$
+ NO$_3^-$) was determined using a bacterial method (Sigman et al., 2001), and gas chromatography (GC;
Thermo Finnigan Gasbench, USA) with a cryogenic extraction and purification system interfaced to an
isotopic ratio mass spectrometer (IRMS; Thermo Fisher Delta V$^{PLUS}$, USA). NO$_X^-$ was quantitatively
converted to N$_2$O using the bacterial strain *Pseudomonas aureofaciens* (ATTC no. 13985). The N$_2$O was
then introduced to the GC-IRMS through the on-line N$_2$O cryogenic extraction and purification system.
The $\delta^{15}$N of NO$_X^-$ was calibrated against nitrate isotope standards (USGS 34, IAEA N3, and USGS 32),
which were run after every 10 samples during the run, as well as before and after each run. Accuracy
(pooled standard deviation) was better than $\pm$0.2‰ based on analyses of these standards at an injection
level of 20 nmol N.
Nitrification rates were primary determined by the accumulation of $^{15}$N in the product pool relative
to the initial conditions using Eq. (1):
$$NR = d[^{15}N_t]/dt \times ([^{14}NH_4^+] + [^{15}NH_4^+])/[^{15}NH_4^+] \qquad (1)$$
where NR is the nitrification rate, t is the incubation time, $[^{15}N_t]$ is the concentration of $^{15}$N in nitrate
plus the nitrite pool in the sample at time t, $[^{14}NH_4^+]$ is the observed natural ammonium concentration
and $[^{15}NH_4^+]$ is the final tracer concentration after the artificial addition of the stable isotope tracer. The
detect limitation of this method is generally better than 0.01 $\mu$mol N L$^{-1}$ d$^{-1}$.

## 2.7 Statistical analysis

Since normal distribution of the individual data sets was not always met, we used the non-parametric

Wilcoxon tests for comparing two variables. Polynomial and exponential growth models (Sigmaplot)

were used to determine the relationships between variables. Canonical correspondence analysis (CCA)

was used to analyze the variations in the nitrifier communities under the constraint of environmental

factors with automatic variable selection procedures in the CANOCO software (version 4.5,

Microcomputer Power, USA) (Ter-Braak, 1989). The gene data were normalized as relative abundances.

The environmental factors were normalized via Z transformation (Magalhães et al., 2008). The null

hypothesis, that the community was independent of environmental parameters, was tested using

constrained ordination with a Monte Carlo permutation test (999 permutations).

The standard and partial Mantel tests, which assess the correlations between two matrices

controlling for the effects of a third matrix, were run in R (VEGAN) to determine the correlations

between environmental factors or nitrification rates and nitrifier population compositions. Dissimilarity

matrices of nitrifier communities were based on Bray-Curtis distances between samples, while

environmental factors and nitrification rates were based on Euclidean distances between samples. The

significance of the Mantel statistics based on Spearman or Kendall's product-moment correlation was

obtained after 999 permutations. The results of the statistical tests were assumed to be significant at

$P$-values $\leq 0.05$.



## 3 Results

### 3.1 Biogeochemical characteristics of the studied transect

According to the geomorphology and geochemical characteristics, the 12 sites in the PRE are situated in

the upper (P1–P6), middle (P7 and P8), and lower reaches (P9–P12) of the estuary (Fig. 1). The upper

reaches receive a small amount of freshwater, sewage, and industrial effluent discharge. The middle

reaches receive about half of the freshwater from the North and West rivers, tributaries of the Pearl

River, with little salinity stratification. The lower reaches are controlled mainly by estuarine mixing of

freshwater and seawater (Wang et al., 2012). Salinity exhibited consistently low values between 0.12

and 3.82 at sites P1–P6 in the PRE upper reaches, but it sharply increased downstream from 1.23 to

31.92 at sites P7–P12 in the middle and lower reaches of the PRE (Fig. 2a). Temperature varied from

26.34 to 30.14°C and decreased seaward (Fig. 2b). Total suspended material concentrations ranged from

1.78 mg $L^{-1}$ in the surface water of site P12 to 100 mg $L^{-1}$ in the bottom water of site P4 (Fig. 2c).

Dissolved oxygen concentrations showed a strong increasing trend seaward from 0.19 to 5.78 mg $L^{-1}$,

with concentrations below 2 mg $L^{-1}$ at sites P1–P6 (Fig. 2d). Accordingly, pH also showed a distinct

increasing trend seaward from 7.04 to 8.17 (Fig. 2e). The nutrient (nitrate/nitrite/ammonium, phosphate,

and silicate) concentrations showed distinctly decreasing trends seaward (Fig. 2f–j). The ammonium

concentrations drastically decreased from 140.1 at site P1 to 9.9 μM at P6 in the upper PRE and had

consistently low concentrations (below detection limit to 16.7 μM) in the middle and lower reaches (Fig.
2f). The nitrite concentrations varied from 1.9 μM in the bottom water (2 m above the seafloor) of site
P12 to 44.2 μM in the bottom water (3.5 m above the seafloor) of site P4 (Fig. 2g). Overall, the upper
PRE was characterized by hypoxic waters containing sufficient nutrients; DO concentrations increased
seaward while the nutrient and TSM concentrations distinctly decreased seaward.

6        Depth profiles of the biogeochemical parameters from SEATS are shown in Fig. S1. Salinity

slightly increased from 32.89 to 34.62 with depth. The sea surface temperature was 28.69°C, while the
temperature decreased sharply to 2.35°C in the deep waters. The ammonium concentrations varied from
below detection limit to 170.75 nM at 140 m depth. The nitrite concentrations ranged from detection
limit to 0.63 μM at 55 m. The nitrate concentrations ranged from below detection limit to 39.32 μM
along the water column. Phosphate and silicate increased from below detection limit to 2.89 μM and
from 2.40 to 145.46 μM, respectively, with increasing water depth.
**3.2 Diversity of ammonia and nitrite-oxidizing microbial communities**
Archaeal and *β*-proteobacterial *amo*A and NOB (*Nitrospira*, *Nitrospina*, and *Nitrobacter*) *nxr*B gene
clone libraries were constructed for the FL communities from the surface and bottom waters at site P8
and P9 because the most dramatic variations in biogeochemical properties along the PRE transect were
present between these two sites (Fig. 2). In addition, archaeal *amo*A gene clone libraries were
constructed at 75, 200, 800, and 3000 m water depth from SEATS, while a NOB *Nitrospina nxr*B gene
clone library was constructed only at 800 m at SEATS as genes were not amplified successfully at the
other three water depths. Rarefaction analyses showed that the diversity of *β*-AOB *amo*A genes
observed in the PRE was nearly exhaustive, while the archaeal *amo*A gene libraries were composed of
more phylotypes in both the PRE and SCS. Moreover, the richness of archaeal *amo*A genes was higher
in the SCS than in the PRE (Fig. S2a). The *nxr*B gene clone libraries might have captured the majority
of *Nitrobacter nxr*B gene types in the PRE with the primer sets used, based on the rarefaction curves,
but not the *Nitrospira* and *Nitrospina nxr*B genes in the PRE and SCS (Fig. S2b). The same conclusions
are supported by the diversity indices (Table S4).
**3.3 Phylogenetic analysis of archaeal *amo*A and *Nitrospira* and *Nitrospina nxr*B genes**
A total of 519 AOA *amo*A gene sequences were recovered and grouped into five clusters (A, Ba, Bb, D,
and E) based on phylogenetic analysis (Fig. 3 and S3). According to the framework of Sintes et al.
(2013) for the Atlantic and Arctic oceans, high ammonia clusters (HAC) were present in environments
where ammonia concentrations ranged from 20 to 100 nM or even higher; however, low ammonia
clusters (LAC) were predominant in environments where ammonia concentrations were frequently
below detection limit. About half of the sequences retrieved from the PRE fell into groups A and D and
almost all sequences retrieved from SEATS fell into groups Ba and Bb. Groups A and D have been
identified as HAC and groups Ba and Bb as LAC by Nunoura et al. (2015) based on a phylogenetic
analysis of archaeal *amo*A genes. Another half of the sequences retrieved from the PRE had an 86% to
100% DNA sequence identity with sequences recovered from high ammonia environments, such as
lakes, rivers, soil, sewage treatment plants, and biofilters and clustered into group E, which clustered
tightly with group D (Fig. 3). Thus, we defined group E as a HAC. The ammonium concentrations at
sites where sequences were recovered further confirmed the categorization of groups A, Ba, Bb, D, and
E. The sequences falling in groups A, D and E (HAC) were retrieved from sites with ammonium
concentrations of 0.032 to 8.09 µM with the exception of four sequences retrieved from 3000 m at
SEATS (below detection limit). The sequences falling in group Ba and Bb (LAC) were retrieved from
SEATS at depths with ammonium concentrations below detection limit, except for 200 m (0.035 µM)
(Fig. 3). Phylogenetic analysis and the relative abundances of each group clearly revealed the distinct
distribution of major *amo*A subgroups from the estuary (HAC) to the SCS central basin (LAC) and from
the upper water (HAC) to the deep ocean (LAC) (Fig. 3 and S3).
A total of 345 *Nitrospira nxr*B gene sequences were recovered. Phylogenetic analysis (Fig. 4)
grouped the sequences into previously described clusters (Pester et al., 2013), except for group H that
only contained sequences recovered from the PRE in this study. Despite containing 95% of all of the
*Nitrospira nxr*B sequences, groups B, C, D and F all belong to *Nitrospira* Lineage II. Notably, group C
was the most dominant branch in the PRE with 92% to 98% DNA sequence identity with *Nitrospira* sp.
enrichment BS10 derived from activated sludge (Spieck et al., 2006). The sequences of group D have
91% to 94% DNA sequence identity with *N. moscoviensis* derived from a heating system (Ehrich et al.,
1995), and the sequences of groups B and F are closely related with the *nxr*B sequences from Austrian
forest soils (Pester et al., 2013). Around 2% of sequences fell into group A, belonging to *Nitrospira*
Lineage I, which could have evolved from an ancestor in *Nitrospira* Lineage II (Pester et al., 2013). The
remaining ~2% of sequences were grouped into groups E (*Nitrospira* Lineage V) and G (*Nitrospira*
Linage IV). *Nitrospira* Linage IV were reported to contain *N. marina* isolated from the Gulf of Maine
(Watson et al., 1986) and sponge-associated *Nitrospira* (Taylor et al., 2007; Off et al., 2010). The *nxr*B
gene of *Nitrospira* was not detected at SEATS.
A total of 185 *Nitrospina nxr*B gene sequences were recovered. The phylogenetic tree grouped the
sequences into four clusters (Fig. 5). The sequences recovered from SEATS all fell into a single branch
(the SCS cluster), which showed high similarity (95% to 99% gene sequence identity) with three
sequences belonging to one OTU from the eastern tropical South Pacific (ETSP) OMZ. The sequences
retrieved from the PRE fell into three other clusters. Around 9% of total sequences clustered in the
ETSP OMZ dominant cluster, and 48% clustered as a unique branch (the PRE cluster), which only
contained sequences obtained from this study. Around 23% of total sequences fell in the 3/211 cluster
with 88% to 100% gene sequence identity with *N. gracilis* 3/211, which was isolated from ocean
surface water (Watson and Waterbury, 1971), and, in this study, was used to design the primers for





amplifying the *nxr*B gene of *Nitrospina*. The phylogenetic analysis and relative abundance of each
group revealed the distinct distribution of major *Nitrospina nxr*B subgroups from the PRE to the SCS
(Fig. 5).
**3.4 Abundance distribution of ammonia and nitrite oxidizers and nitrification rates**
Abundances of the archaeal and *β*-proteobacterial *amo*A genes and *Nitrospira* and *Nitrospina* 16S rRNA
genes were quantified using the qPCR method at all 12 sites of the PRE for the FL and PA communities
in the surface and bottom waters (Table S1). *Nitrobacter* and *Nitrococcus* were not quantified since they
were not major NOB groups in either the PRE or SCS sites, as indicated by clone library analysis.
Archaeal and *β*-proteobacterial *amo*A gene abundances varied from below detection limit to $4.54 \times 10^5$
copies $L^{-1}$ (PA community in the bottom water of site P9) and from below detection limit to $3.42 \times 10^4$
copies $L^{-1}$ (PA community in the bottom water of site P4), respectively. Overall, the archaeal *amo*A
genes were significantly more abundant than the *β*-proteobacterial *amo*A genes (Wilcoxon, $P <0.01$),
but AOB more distinctly attached to particles compared with AOA in the upper reaches of the PRE
(sites P1–P6; Fig. 6a and b). *Nitrospira* and *Nitrospina* 16S rRNA gene abundances varied from below
detection limit to $2.02 \times 10^6$ copies $L^{-1}$ (PA community in the bottom water of site P4) and from 51 to
$3.81 \times 10^5$ copies $L^{-1}$ (PA community in the bottom water of site P4), respectively. The *Nitrospira* 16S
rRNA genes were significantly more abundant than the *Nitrospina* 16S rRNA genes in the upper and



middle reaches of the PRE (sites P1–P8, Wilcoxon, $P < 0.01$), whereas the opposite trend was observed
in the lower estuary (sites P9–P12, Wilcoxon, $P < 0.01$; Fig. 6c and d). All of the genes were
significantly more abundant in the PA than the FL communities (Wilcoxon, $P < 0.01$) (Fig. 6e and f).

4        Sites P1–P6, located in hypoxic waters that are typically defined when DO concentrations fall

below 2 mg $L^{-1}$ (Renaud, 1986), of the PRE upper reaches, have DO concentrations ranging from 0.19
to 1.93 mg $L^{-1}$ (Fig. 7). Generally, the abundance of NOB (sum of *Nitrospira* and *Nitrospina*) 16S rRNA
genes was significantly higher than the ammonia-oxidizing microbes (AOM, sum of archaea and
*β*-proteobacteria) *amo*A genes in the hypoxic waters (Wilcoxon, $P < 0.01$; Fig. 6g and h). Notably,
significant positive relationships were observed between AOM and NOB groups for both the FL (Fig.
8a) and PA (Fig. 8b) communities (eight correlations, $P < 0.05$–0.01, the findings were the same
excluding the maximum values), suggesting a coupling between ammonia and nitrite oxidizers in the
hypoxic estuarine niche.

13       The hypoxic zone gradually disappears seaward and the DO concentrations of sites P7–P12 varied

from 2.15 to 5.78 mg $L^{-1}$ (Fig. 7). The significant relationship between AOM and NOB collapsed
instantly. The abundance of the NOB 16S rRNA genes was comparable with the AOM *amo*A genes (Fig.
6g and h), and archaea and *Nitrospina* became the dominant ammonia and nitrite oxidizers, respectively
(Fig. 6b and d–f).

18       The nitrification rates generally decreased seaward with increasing DO concentrations, ranging



from 0.19 µmol L$^{-1}$ day$^{-1}$ in the bottom water (2 m above the seafloor) of site P9 to 75.81 µmol L$^{-1}$
day$^{-1}$ in the bottom water (3.5 m above the seafloor) of site P5 (Fig. 7). Distinctly higher nitrification
rates were observed in the hypoxic zone than the middle and lower reaches of the PRE (Wilcoxon
rank-sum test, $P < 0.05$).

## 6 4 Discussion

### 7 4.1 Coverage of the primer pair for *Nitrospina nxr*B genes

The primer pair of nxrBNF and nxrBNR targeting the *Nitrospina nxr*B genes was designed in this study
according to two *nxr*B gene sequences of *N. gracilis* 3/211, which is the only isolated *Nitrospina* strain
from the oxygenated ocean (Watson and Waterbury, 1971) and the only genome-sequenced *Nitrospina*
so far (Lücker et al., 2013). Despite very few reference sequences, phylogenetic analysis of the
*Nitrospina nxr*B gene sequences retrieved based on this primer pair indicated diverse phylogenetic taxa,
including 12 OTUs and four major phylogenetic clusters. The relative abundances of the four groups
showed that 77% of total sequences fell out of the 3/211 cluster (Fig. 5). Feng et al. (2016) and Rani et
al. (2017) also designed primer pairs targeting *nxr*B and *nxr*A subunit genes of *Nitrospina*, respectively.
However, Feng et al. (2016) did not obtain any *nxr*B target fragments and Rani et al. (2017) focused on
the *nxr*A gene in marine sediments.



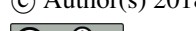

**4.2 Coupling between ammonia and nitrite oxidizers in the estuarine hypoxic niche**

The abundance of NOB 16S rRNA genes was significantly higher than the AOM *amo*A gene in PRE

hypoxic waters. This is similar to previous observations that NOB can reach high abundances in oceanic

OMZs, where *Nitrospina* and *Nitrococcus* are abundant (Füssel et al., 2012; Beman et al., 2013).

However, in PRE hypoxic waters, *Nitrospira* and *Nitrospina* were dominant, particularly on the

particles. Taken together, distinctly higher nitrification rates in the hypoxic zone and extremely low

oxygen concentrations suggests that the PRE system could not supply oxygen fast enough to meet the

demands of NOB and thus oxygen may not be the only electron acceptor. It was hypothesized that

abundant NOB in a hypoxic zone might benefit from utilizing alternative terminal electron acceptors for

nitrite oxidation, such as iodate, Mn(IV) or Fe(III) (Lam and Kuypers, 2011; Casciotti and Buchwald,

2012), which could be more reactive in the particles in hypoxic waters (Hsiao et al., 2014).

Significant positive relationships between AOM and NOB groups in the PRE hypoxic waters for

both PA and FL communities suggest a coupling between ammonia and nitrite oxidizers. Similar

observations were also found by Mincer et al. (2007) and Santoro et al. (2010) where the distribution

profiles of total AOA and *Nitrospina* were correlated in some coastal and open ocean habitats. In

Namibian soils, network analysis also indicated that AOA and *Nitrospira* communities were highly

correlated (Pester et al., 2013). The tight coupling between ammonia and nitrite oxidizers in abundance

and spatial distribution, known as the "nitrification aggregate" (Arp and Bottomley, 2006), could reflect

their interactions (Daebeler et al., 2014). The reciprocal feeding (Daims et al., 2016) supports such
interactions between nitrifiers. For example, urease-positive (Koch et al., 2015) or cyanase-positive
(Starkenburg et al., 2006; Lücker et al., 2010; 2013; Palatinszky et al., 2015) NOB can provide AOM
with ammonia from urea and cyanate degradation while NOB obtain nitrite from the AOM. In high
particle load environments, such reciprocal feeding interactions might be more prominent than in the
open ocean because particles, as well as sludge flocs or biofilms, could provide matrices for the
complex interactions of these nitrifiers.
**4.3 Succession of dominant nitrifier groups from the estuary to the open ocean**
Although the archaeal *amo*A genes were generally more abundant than the $\beta$-AOB *amo*A genes,
significant positive correlations were observed between the $\beta$-AOB *amo*A gene abundance and the
nitrification rate (oxidation of ammonia to nitrate) in the PRE ($r = 0.785$, $P < 0.05$; the partial Mantel
test controlling for the effects of the NOB abundance: $R = 0.786$, $P < 0.01$). This result suggests that
AOB might be more active than AOA, prefer estuarine habitats, and thus dominate the nitrification rate.
AOA have been detected in great numbers in coastal and estuarine waters, such as the Columbia River
estuary, Monterey Bay, Southern California Bight, San Francisco Bay, Yangtze River estuary and Bering
Strait (Crump et al., 2000; Mincer et al., 2007; Beman et al., 2008; Mosier et al., 2008; Zhang et al.,
2014a; Damashek et al., 2017), while AOB often comprise less than 0.1% of the microbial community



(Bothe et al., 2000). However, high abundance does not necessarily indicate high turnover rates (Zhang
et al., 2014b) and AOB in ammonium-enriched environments might be highly active (Füssel, 2014) and
thus substantially contribute to ammonia oxidation despite their low abundance. Similarly, the *β*-AOB
*amo*A gene abundances have been correlated with potential nitrifying activities in the waters of the
Seine River estuary (Cébron et al., 2003). In addition, the *β*-AOB *amo*A gene abundances were found to
be significantly correlated to more environmental factors, including nitrite, nitrate, silicate, salinity,
TSM, DO, and pH, in the PRE, whereas only one factor (TSM) was correlated to the AOA *amo*A gene
(Table S5). We speculate that AOB could be better adapted to the estuarine habitat than AOA.
*Nitrospira* was more abundant than *Nitrospina* in the upper and middle reaches of the PRE.
Moreover, a significant positive correlation was observed between the *Nitrospira* 16S rRNA gene
abundance and the nitrification rate in the PRE ($r = 0.791$, $P < 0.05$; the partial Mantel test controlling
for the effects of the *amo*A gene abundance: $R = 0.163$, $P < 0.05$). These results suggest that *Nitrospira*
could be well adapted to eutrophic estuarine environments, with both higher abundance and nitrifying
potential. *Nitrospira* is widespread in diverse habitat types and especially abundant in freshwater (Koch
et al., 2015) and estuarine (Cébron et al., 2005; Nakamura et al., 2006) environments, but less abundant
in marine ecosystems (Hoffmann et al., 2009; Off et al., 2010) despite the fact that the first *Nitrospira*
described was isolated from an ocean (Watson et al., 1986).
Archaea and *Nitrospina* became the dominant ammonia and nitrite oxidizers, respectively, along
the transect from the PRE to the SCS. This succession of dominant groups can be explained by niche
differentiation of these nitrifiers, which involves different adaptations to environmental parameters,
ecological strategies, and microbe–microbe interactions. For instance, AOB and *Nitrospira* might be
enriched on particles or aggregates (Phillips et al., 1999; Lam et al., 2004; Lebedeva et al., 2008;
Haaijer et al., 2013; Ganesh et al., 2014; Zhang et al., 2014a) and play an important role in estuarine
ecosystems characterized by high particle densities, whereas AOA and *Nitrospina* might be relatively
more adaptable to a FL life strategy (Watson and Waterbury, 1971; Woebken et al., 2007; Ganesh et al.,
2014) and thus abundant in low-particle environments.
**4.4 Environmental parameters allowing niche differentiation**
The CCA analysis based on qPCR data (Fig. 9) revealed that AOB and *Nitrospira* were more adaptable
to high nutrient and TSM concentrations; in contrast, AOA and *Nitrospina* FL communities were more
adaptable to high salinity, DO, and pH water masses and low nutrient and TSM environments. To some
extent, AOA and *Nitrospina* PA communities were positively influenced by TSM. The CCA analysis
based on clone libraries (Fig. 10a) further revealed that AOA HAC groups E and D were under the
constraint of high nutrient conditions and HAC group A was positively influenced by TSM to an extent.
The LAC groups Ba and Bb were under the constraint of high salinity and low temperature water
masses. This is consistent with the phylogenetic analysis that indicates niche differentiation of AOA



subgroups by adaptation to different ammonia levels. Similarly, the *Nitrospina* SCS cluster was under
the constraint of high salinity and low temperature water masses, and other clusters were positively
correlated with nutrients or TSM (Fig. 10b). The *Nitrospira* OTU-based ordination was obviously
correlated with nutrients, DO, TSM, and salinity in the PRE. Overall, groups d and g were positively
correlated with salinity and TSM, and other groups were regulated by nutrients and DO (Fig. 10c).
Taken together, these CCA analyses show how environmental parameters allow for the niche
differentiation of these nitrifiers.

8       The environmental factors included three types: water mass parameters (temperature, salinity, and

silicate), substrate parameters (ammonia/ammonium, nitrite, and nitrate), and parameters influencing
substrate availability (DO, TSM, and pH). AOA have been shown to be adaptable to low ammonia
concentrations (<10 nM ammonium threshold, $K_{m(app)}$ = ~3 nM $NH_3$; Martens-Habbena et al., 2009;
Kits et al., 2017), whereas AOB require higher concentrations of ammonia than usually observed in the
ocean ($K_{m(app)}$ = 0.25–157.50 μM $NH_3$; Kits et al., 2017). Therefore, AOA are the major ammonia
oxidizers in estuarine, coastal, and oceanic environments (Francis et al., 2005; Lam et al., 2007; Beman
et al., 2008; Santoro et al., 2010), and AOB are favored in high ammonium environments (Verhamme et
al., 2011). Furthermore, the niche differentiation of AOA subgroups also show their adaptation to
different ammonia levels.
Nitrite, a central intermediate compound in nitrification, was positively correlated to NOB 16S
rRNA and *β*-proteobacterial *amo*A gene abundances ($P$ <0.05–0.01, Table S5). *Nitrospira* displays
stronger correlations to nitrite than *Nitrospina* in the PRE, suggesting that *Nitrospira* is likely adapted to
a higher nitrite flux (Spieck et al., 2006; Lebedeva et al., 2008; Nunoura et al., 2015). Nitrite might be
one major factor causing niche differentiation of NOB groups (Both and Laanbroek, 1991). Nitrate, a
final product of nitrification, was also significantly positively correlated to *Nitrospira* 16S rRNA and
*β*-proteobacterial *amo*A gene abundances ($P$ <0.05–0.01, Table S5).

7        Notably, all genes were significantly positively correlated to TSM concentrations in PA and total

communities ($P$ <0.05–0.01, Table S5). This is consistent with the observation that all of the genes were
significantly more abundant in the PA communities. The suspended particulate microniche could be
beneficial to microbial activity because of the vicinal supply of nutrients or substrates from particles
(Belser, 1979; Crump et al., 1998; Ouverney and Fuhrman, 2000; Teira et al., 2006; Zhang et al., 2014a).
Lower light inhibition could also be a potential reason because of particle protection. The DO
concentrations showed a significant negative correlation to the *β*-AOB *amo*A and *Nitrospira* 16S rRNA
gene abundances ($P$ <0.05, Table S5). Previous studies have shown that ammonia oxidizers are highly
abundant under low oxygen conditions because of relatively high ammonia levels (Lam et al., 2007;
Beman et al., 2008; Park et al., 2010; Yan et al., 2012), which might benefit the activity of AOB.
Accumulations of nitrite under low oxygen conditions would also help NOB *Nitrospira* to oxidize
nitrite (Füssel et al., 2012; Beman et al., 2013). pH was also negatively correlated to the *β*-AOB *amo*A
and *Nitrospira* 16S rRNA gene abundances ($P$ <0.05–0.01, Table S5). This is not consistent with
previous studies that showed AOA and AOB *amo*A gene abundances increasing with pH in soils
(Gubry-Rangin et al., 2011), sediments (Rani et al., 2017), and the open ocean (Nunoura et al., 2015). It
is possible this is related to lower availability of the substrate (ammonia) due to increased ionization to
ammonium as pH decreases. However, in an estuary with sufficient nutrients, such as the PRE, negative
correlations between gene abundances and pH could in fact be attributed to co-varying of pH with DO
concentrations.

8        In estuarine ecosystems, water mass mixing highly influences the distribution of microbial

populations. Both silicate and salinity have been previously recognized as one of the most common
indicators to discriminate river water sources in the ocean (Moore, 1986). In this study, silicate
concentrations and salinity were found to be positively and negatively correlated, respectively, to the
*β*-AOB *amo*A and *Nitrospira* 16S rRNA gene abundances ($P$ <0.05–0.01, Table S5), suggesting that
*β*-AOB and *Nitrospira* recovered in the PRE could partly originate from the Pearl River or upstream.

14       Partial Mantel tests were further applied to the qPCR dataset and environmental parameters to

eliminate the co-varying effect of water mass and substrate availability, and to identify the major
process that influences the nitrifier distribution from the estuary to open ocean (Fig. 11). Variations in
the distribution of nitrifier populations along the transect were significantly correlated with water mass
mixing and substrate availability (standard and partial Mantel tests, $P$ <0.05–0.01), except that



ammonia-oxidizing populations only correlated to water mass properties (Fig. 11a–i). Notably, however, water mass parameters and those influencing substrate availability significantly controlled variations in the distribution of FL and PA nitrifier populations along the transect (standard and partial Mantel tests, $P$ <0.05–0.01, Fig. 11j–o). This suggests that nitrifiers' life strategies to some extent allow them to be adaptable to substrate availability.

## 5 Summary

Our work explored the niche differentiation of main nitrifier groups (AOA, $\beta$-AOB, NOB *Nitrospira* and *Nitrospina*) from an estuary (PRE) to the open ocean (SCS), and investigated possible environmental parameters allowing this niche differentiation. These environmental factors included water mass parameters (temperature, salinity, and silicate), substrate parameters (ammonia/ammonium, nitrite, and nitrate), and parameters influencing substrate availability (DO, TSM, and pH). We showed that, from the PRE to the SCS, niche differentiation of nitrifier populations is primarily regulated by water mass mixing and the availability of electron donors (substrate availability). Additionally, the nitrifier populations might have specific adaptations to different substrate conditions provided through their ecological/life strategies (e.g. particle-attached). Therefore, the abundance and activity of nitrifiers could reflect a possible substrate, e.g. ammonia/ammonium or nitrite, flux/availability in ecosystems, providing a biogeochemical clue for understanding carbon and nitrogen cycles.


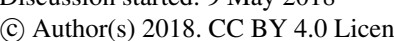


## Data availability

The sequences used for this study were deposited in GenBank under accession numbers KY387947–KY388465 and MG025956–MG026485. The qPCR data were available within this paper (Table S1). Other data can be accessed in the form of Excel spreadsheets via the corresponding author.

**The Supplement related to this article is available online.**

## Author contribution

Y.Z. conceived and designed the experiments. L.H., X.X., and X.W. performed the experiments. L.H., X.X., Y.Z., and X.W. analysed the data. Y.Z., L.H., and X.X. wrote the paper. X.W., S.J.K., and N.J. contributed to the interpretation of results and critical revision.

## Competing interests

The authors declare no conflicts of interest.

## Acknowledgments

We thank Professor Minhan Dai for providing the sampling opportunity during the PRE cruise and





nutrient data. We also thank Zuhui Zuo, Zhuoyu Chen, and Duo Zhao for their assistance in DNA/RNA
extraction and qPCR measurements. This work was funded by the National Key Research and
Development Programs (2016YFA0601400) and NSFC projects (41676125, 41721005, and 91428308).
This study is a contribution to the international IMBER project. We thank Kara Bogus, PhD, from
Liwen Bianji, Edanz Editing China ([www.liwenbianji.cn/ac](www.liwenbianji.cn/ac)), for editing the English text of a draft of
this manuscript.

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



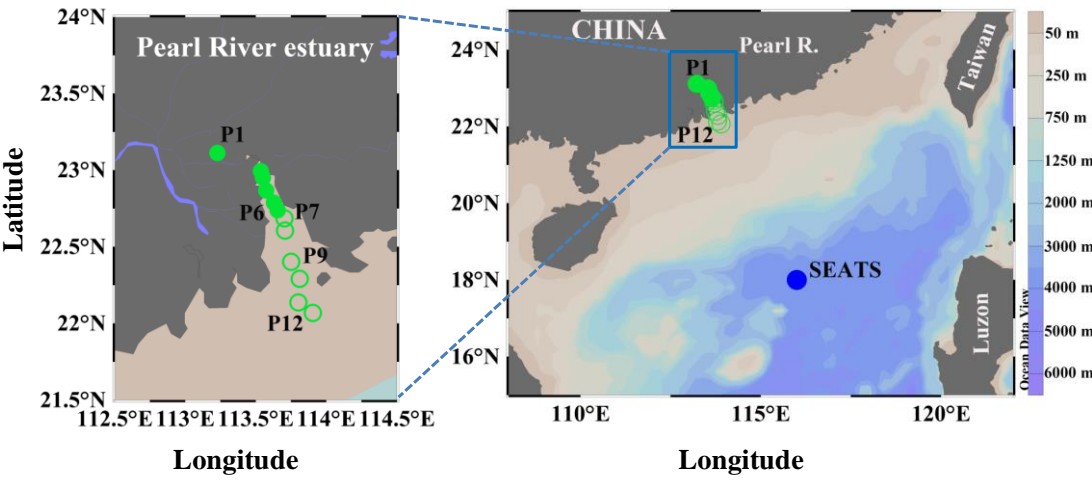

**Figure 1. Site locations and bathymetry.** The solid green circles indicate hypoxic sites in the PRE,

open green circles indicate (low) oxygenated sites in the PRE, and the solid blue circle indicates SEATS

in the central basin of the SCS. This figure was produced using Ocean Data View v. 4.6.2

(http://odv.awi.de, 2014). Isobaths are regarded as the background and the color bar indicates depth.







**Figure 2. Distributions of biogeochemical factors along the PRE transect.** (a) Salinity, (b) temperature, (c) TSM, (d) DO, (e) pH, (f) ammonium, (g) nitrite, (h) nitrate, (i) phosphate, and (j)





1    silicate concentration. P1−12 indicate PRE sampling sites. Black dots indicate sampling depths.







**Figure 3. Unrooted neighbor-joining (NJ) phylogenetic tree of the archaeal *amo*A gene sequences.**
Clone sequences from this study are shown in bold and sequences sharing 95% DNA identity are
grouped. GenBank accession numbers are shown. Groups A, Ba, Bb, and D were defined in Nunoura et
al. (2015) and group E is defined in this study. The relative abundance of clones retrieved for each
library in the five subgroups is indicated by a bar. Total number of clones for each library is shown in
parentheses. Location of sites P8 and P9 (S and B indicate surface and bottom waters, respectively) and
SEATS (S) are shown in Fig. 1. Ammonium concentrations are shown in square brackets. Phylogenetic
relationships were bootstrapped 1000 times, and bootstrap values greater than 50% are shown. The scale
bar indicates 5% estimated sequence divergence. HAC, high ammonia cluster; LAC, low ammonia
cluster. N, not measured; B, below detection limit.



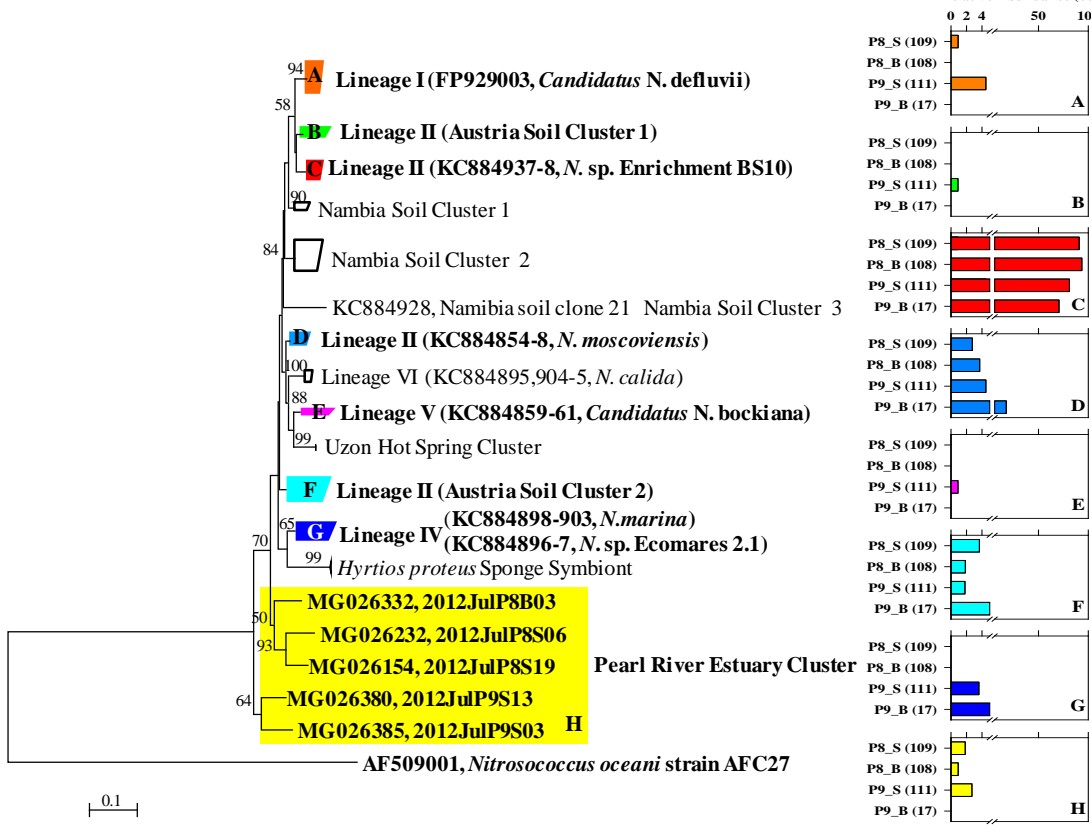

**Figure 4. Rooted neighbor-joining (NJ) phylogenetic tree of the *Nitrospira nxr*B gene sequences.**

Clone sequences from this study are shown in bold and sequences sharing 95% DNA identity are

grouped. GenBank accession numbers are shown. Groups a, b, c, d, e, f, and g are defined according to

Pester et al. (2013), and Group h (highlighted in yellow) is defined in this study. The relative abundance

of clones retrieved for each library in the eight subgroups is indicated by a bar. Total number of clones

for each library is shown in parentheses. Location of sites P8 and P9 (S and B indicate surface and

bottom waters, respectively) are shown in Fig. 1. Phylogenetic relationships were bootstrapped 1000

times, and bootstrap values greater than 50% are shown. The scale bar indicates 10% estimated

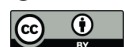

1    sequence divergence.




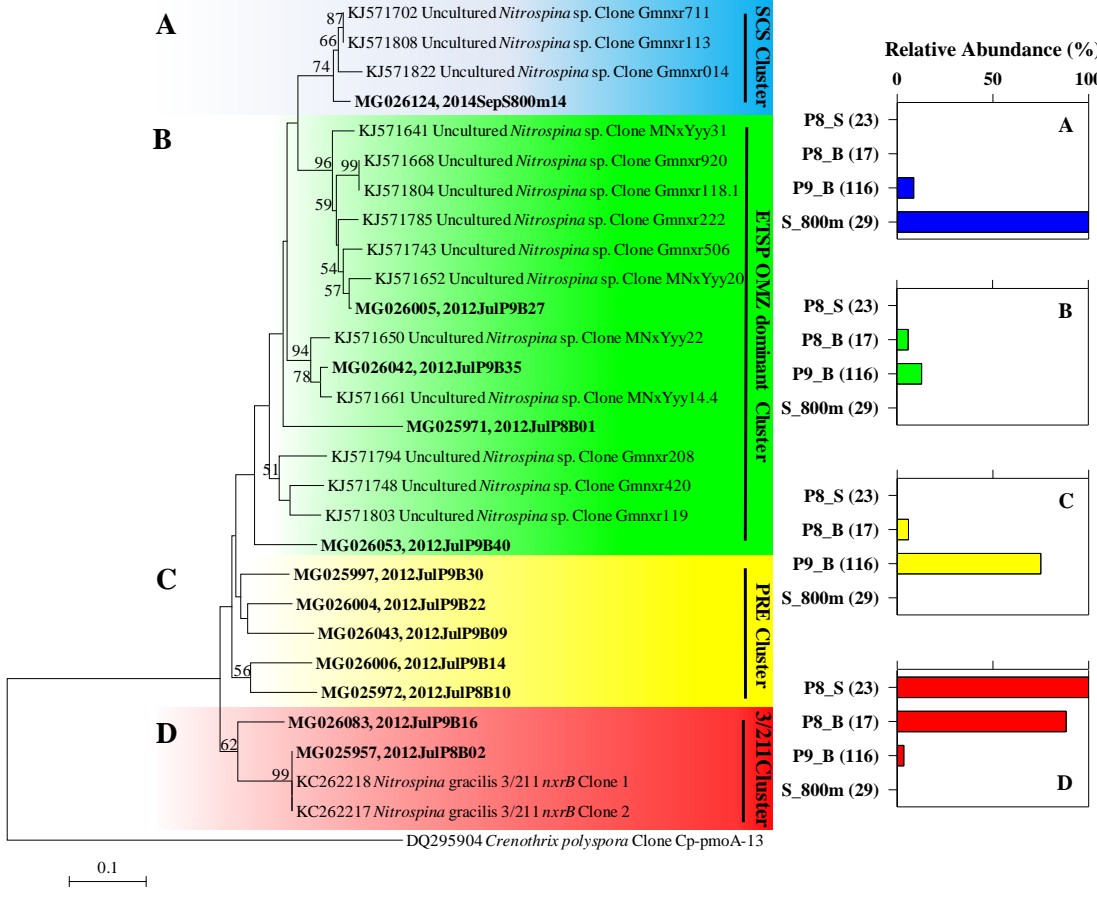

**Figure 5. Rooted neighbor-joining (NJ) phylogenetic tree of the *Nitrospina nxr*B gene sequences.**

Clone sequences from this study are shown in bold and sequences sharing 95% DNA identity are

grouped. GenBank accession numbers are shown. Groups A, B, C, and D are defined in this study. The

relative abundance of clones retrieved for each library in the four subgroups is indicated by a bar. Total

number of clones for each library is shown in parentheses. Location of sites P8 and P9 (S and B indicate

surface and bottom waters, respectively) and SEATS (S) are shown in Fig. 1. Phylogenetic relationships

were bootstrapped 1000 times, and bootstrap values greater than 50% are shown. The scale bar indicates





1    10% estimated sequence divergence.





**Figure 6. Gene abundance distribution of four nitrifier groups along the PRE transect.** (a)

Relative abundance of archaeal (AOA) and $\beta$-proteobacterial (AOB) *amo*A genes in total FL AOM





(sum of archaea and *β*-proteobacteria) *amo*A genes. (b) Relative abundance of AOA and AOB *amo*A
genes in total PA AOM *amo*A genes. (c) Relative abundance of *Nitrospira* and *Nitrospina* 16S rRNA
genes in total FL NOB (sum of *Nitrospira* and *Nitrospina*) 16S rRNA genes. (d) Relative abundance of
*Nitrospira* and *Nitrospina* 16S rRNA genes in total PA NOB 16S rRNA genes. (e) Relative abundance
of FL and PA AOM *amo*A genes in total *amo*A genes. (f) Relative abundance of FL and PA NOB 16S
rRNA genes in total 16S rRNA genes. (g) Relative abundance of AOM *amo*A and NOB 16S rRNA
genes in total FL nitrifier genes. (h) Relative abundance of AOM *amo*A and NOB 16S rRNA genes in
total PA nitrifier genes. Depth-weighted abundances were used to calculate relative abundances for each
site. B, only the bottom water was sampled; S, only the surface water was sampled.



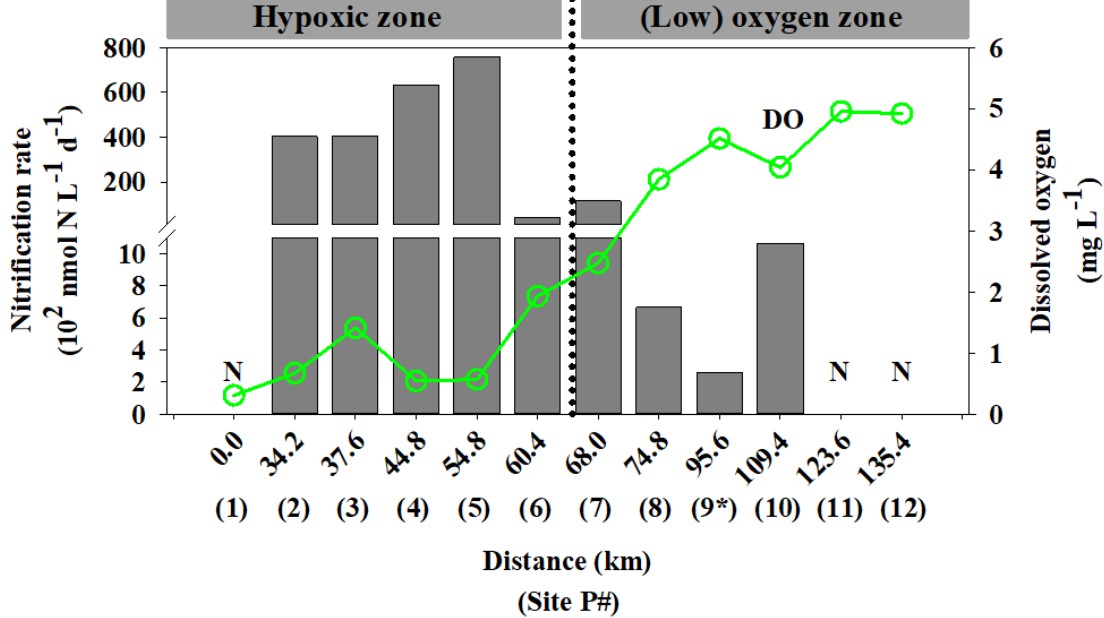

**Figure 7. Nitrification rates and DO concentrations along the PRE transect.** Nitrification rates were

only measured in the bottom waters except for site P9, where rates were measured in both surface and

bottom waters. N, not measured; *the depth-weighted value was used.



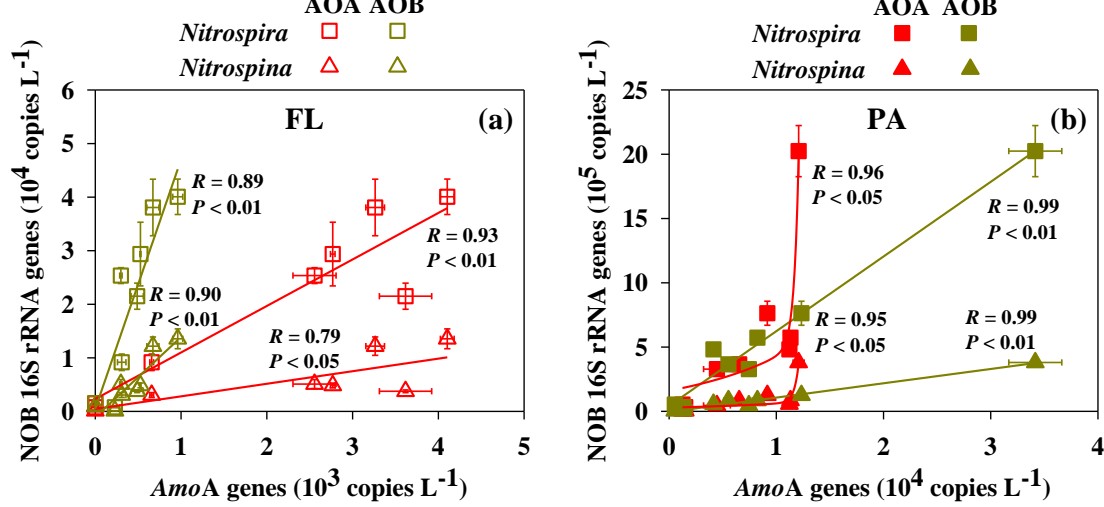

**Figure 8. Correlations between ammonia and nitrite oxidizers in the hypoxic zone of the PRE (sites P1–6).** There are significant positive correlations (n = 8) between archaeal and $\beta$-proteobacterial *amo*A genes and *Nitrospira* and *Nitrospina* 16S rRNA gene abundances in (a) FL and (b) PA communities. Error bars represent standard deviations.





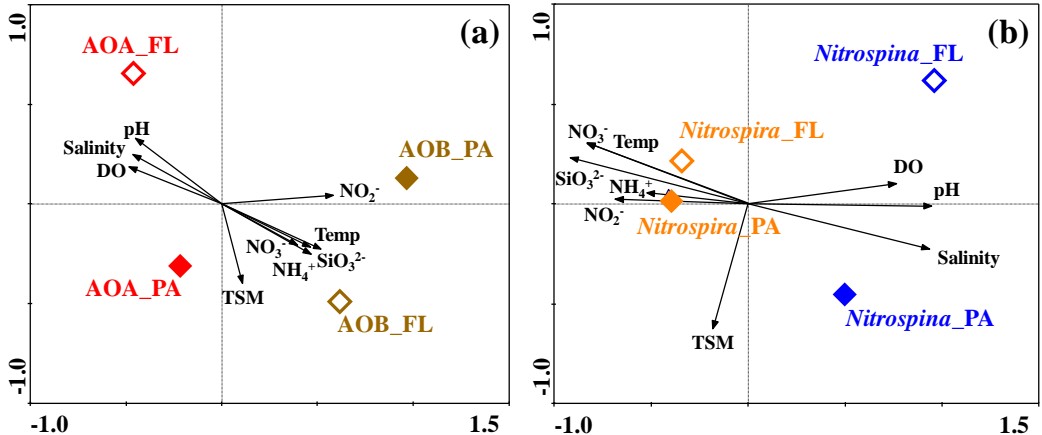

**Figure 9. Canonical correspondence analysis.** (a) Ammonia and (b) nitrite oxidizers under the

constraint of environmental factors. Each diamond represents an individual subgroup. Vectors represent

the environmental variables. Temp, temperature.



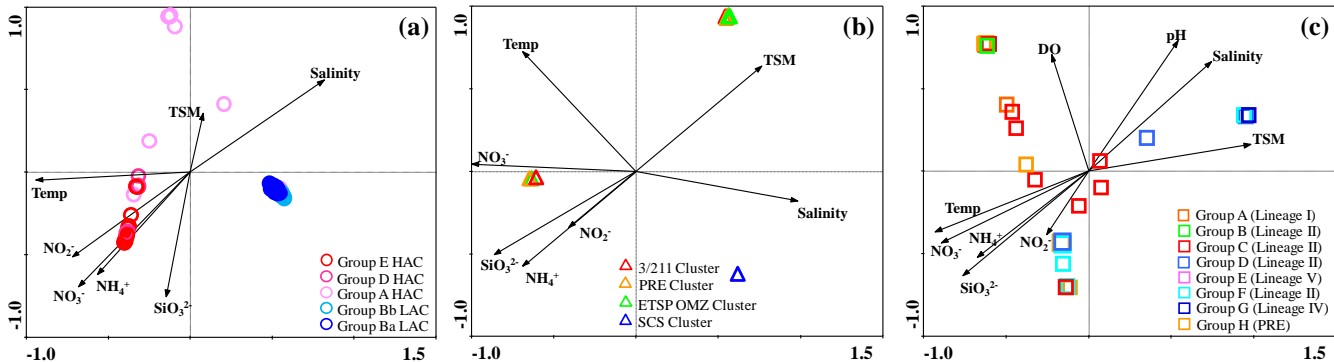

**Figure 10. Canonical correspondence analysis (CCA).** (a) Ammonia-oxidizing archaea, (b) *Nitrospina*, and (c)

*Nitrospira* phylogenetic taxa under the constraint of environmental factors. Each symbol represents an individual OTU.

Vectors represent the environmental variables. Temp, temperature. DO and pH were not included in (a) and (b) because

they were not measured at SEATS.











**Figure 11. Correlations between nitrifier community composition and water mass parameters (temperature, salinity, and silicate), substrate parameters (ammonia/ammonium, nitrite, and nitrate), or parameters influencing substrate availability (TSM, DO, and pH).** Standard and partial Mantel tests were run to measure the correlation between two matrices. Dissimilarity matrices of nitrifier communities were based on Bray-Curtis distances; environmental factors were based on Euclidean distances between samples. Spearman or Kendall's correlation coefficient ($R$) values are shown for standard (first value) and partial Mantel (second and third) tests. The $P$-values were calculated using the distribution of the Mantel test statistics estimated from 999 permutations. $^{*}P < 0.05$; $^{**}P < 0.01$. Matrix of the nitrifier community was calculated according to (a–c) ammonia-oxidizing archaeal and bacterial abundances (AOB vs. AOA), (d–f) *Nitrospira* and *Nitrospina* abundances (*Nitrospira* vs. *Nitrospina*), (g–i) ammonia and nitrite-oxidizing microbial abundance (AOM vs. NOB), (j–l) FL and PA ammonia-oxidizing archaeal and bacterial abundances (AOB vs. AOA, FL vs. PA), and (m–o) FL and PA *Nitrospira* and *Nitrospina* abundances (*Nitrospira* vs. *Nitrospina*, FL vs. PA). (a, d, g, j, and m) Matrix of substrate parameters included $NH_4^+$, $NO_2^-$, and $NO_3^-$ concentrations, (b, e, h, k, and n) matrix of water mass parameters included temperature (Temp), salinity, and $SiO_3^{2-}$, and (c, f, i, l, and o) matrix of parameters influencing substrate availability included TSM, DO, and pH.