# Peer review of "Niche differentiation of ammonia and nitrite oxidizers along a salinity gradient from the Pearl River estuary to the South China Sea"

_Biogeosciences, 2018_

## Short Comment (SC1) · 25 May 2018

The authors characterized the diversity, abundance and activity of nitrifiers associated with waters from the Pearl River estuary to the South China Sea. The data set provides novel insights into the niche separation and interactions of ammonia and nitrite oxidizers. However, I found technical issues in the quantification of archaeal ammonia oxidizers in the current version of manuscript as shown below.

In this study, archaeal amoA genes were used for a molecular marker for archaeal ammonia oxidizers, and there are technical issues in both diversity analysis and quantitative PCR. 1. The composition of archaeal amoA diversity is highly biased in PCR

amplification comparing to the SSU rRNA gene analysis as reported previously (Meinhardt et al., 2015; Nunoura et al., 2015). In addition, it would lead inappropriate selection of sequences used to obtain standard curves in qPCR. 2. Thus, the authors should mention about the possibility that the interpretation of the niche separation among AOA subgroups may be influenced by PCR bias in this manuscript. 3. The authors obtained archaeal amoA gene sequences in the clone analysis and then used the selected sequences to obtain standard curves in the following qPCR. However, qPCR primer set for archaeal amoA genes was identical to a primer set used for the conventional clone analysis. This generally allows complete match between archaeal amoA clone sequences and primer sequences in PCR reaction for obtaining standard curves. In contrast, the presence of few mismatch residues is expected between environmental amoA gene sequences and primer sequences in qPCR. The gap would be a reason for underestimation of archaeal amoA genes in environmental samples. In qPCR of archaeal amoA, a primer set, Wuchter et al. 2006, or other primer set that does not overlap the annealing regions in the initial clone analysis is recommended.

Specific comments P2, L3: We analyzed diversity and abundance of ammonia-oxidizing archaea (AOA) and betaproteobacteria (AOB), nitrite-oxidizing bacteria (NOB), and nitrification rates to

P2, L6-7: AOA were generally more abundant than betaproteobacterial AOB, however,

P2, L12: What does "a coupling of ammonia and nitrite oxidizers" mean?

P4, L2-14: Please insert a sentence to present the close relationship between Nitrospina and "Ca. Nitromaritima". Sequences belong to "Ca. Nitromaritima" had been reported as a group in Nitrospina until the definition of "Ca. Nitromaritima". Thus, the discussion in Lucker et al. 2013 includes both Nitrospina and "Ca. Nitromaritima".

P4, L6: Information from Pachiadaki et al. 2017 should be referred through the manuscript.
P6, L5: Strains and/or genomic DNA from the public repositories used in this study should be summarized as the first paragraph in Materials and methods.

P8, L18-: Did the authors determine OTUs for each library or among the libraries obtained in this study?

P8, L18-: Did the authors conduct any chimera check programs? It has been reported that more than 10% of the archaeal amoA gene sequences in the public database are chimera sequences (Eloy Alves et al. 2018).

P9, L13: Names of the sequence used to obtain standard curves for the qPCR should be presented.

P9, L14: How did the authors obtain DNA fragments?

P11, L5: Pseudomonas chlororaphis subsp. aureofaciens (ATCC 13985)

P13, L3-: Did the authors obtain data of turbidity or light intensity during the sampling?

P15, L11-: Please clarify how many clone libraries constructed for each gene. Supplementary tables presenting distribution of OTUs will help readers to understand the results.

P18, L10: Please present the values of detection limits in each qPCR if possible.

P19, L6: As I know, the abundance of ammonia oxidizers is generally higher than nitrite oxidizers in aquatic environments. I am afraid that the result was influenced by the technical issues described above.

P21, L5: dominant NOB.

P21, L5-: Information from Hawley et al. 2014 should be integrated in this discussion.

P25, L17: "availability of ammonia" or "ammonia concentration/flux" would be better than "ammonia levels".

P25, L17: Appropriate references should be provided.

P26, L12: Appropriate references for light inhibition on the growth of nitrifiers should be provided.

References: Proc. Natl. Acad. Sci. USA instead of P. Natl. Acad. Sci. USA

Fig. 5: Did the Nitromaritima sequence excluded in this phylogenetic analysis?

---

## Author Comment (AC1) · 17 Jun 2018

Manuscript Number: bg-2018-189

Manuscript title: Niche differentiation of ammonia and nitrite oxidizers along a salinity gradient from the Pearl River estuary to the South China Sea

Response to Reviewer #1

We greatly thank the reviewer for the valuable comments, useful suggestions and careful revisions, based on which we have revised the manuscript. And the point-by-point responses to the comments are shown below.

[Figure]

[Figure]

T. NUNOURA

takuron@jamstec.go.jp

The authors characterized the diversity, abundance and activity of nitrifiers associated with waters from the Pearl River estuary to the South China Sea. The data set provides novel insights into the niche separation and interactions of ammonia and nitrite oxidizers. However, I found technical issues in the quantification of archaeal ammonia oxidizers in the current version of manuscript as shown below. In this study, archaeal amoA genes were used for a molecular marker for archaeal ammonia oxidizers, and there are technical issues in both diversity analysis and quantitative PCR. 1. The composition of archaeal amoA diversity is highly biased in PCR amplification comparing to the SSU rRNA gene analysis as reported previously (Meinhardt et al., 2015; Nunoura et al., 2015). In addition, it would lead inappropriate selection of sequences used to obtain standard curves in qPCR.

Response: Meinhardt et al. (2015) showed that the archaeal amoA diversity retrieved from soil using their new-designed GenAOA primer set was more similar to that from the metagenomic data, compared with using FranAOA primer set, which was used in our manuscript. However, all the mainly different clades (Nitrosotalea 1.1, Nitrososphaera 1 (non-54d9), Nitrososphaera 1.1 and Nitrososphaera 54d9) between FranAOA clone library and metagenomic data were not adapted to the marine habitats.

Nunoura et al. (2015) showed that the distribution and abundance patterns of four subgroups of archaeal amoA genes (Group A, Ba, Bb, and D) in the Challenger Deep measured by qPCR were more similar to the MGI SSU rRNA gene clone library community structure, rather than the archaeal amoA gene clone library (using the FranAOA primer set). However, more specifically, above the Abyssal zone (4000-6000 m), the qPCR abundance patterns of the archaeal amoA subgroups were similar to the archaeal amoA gene clone library analysis; the inconsistent pattern was present in the

hadal zone (below 6000 m). Our clone library analysis of archaeal amoA gene was carried out at two estuary sites and in the South China Sea basin at 75 m, 200 m, 800 m, and 3000 m water depth. Our qPCR measurements of archaeal amoA gene were only performed in the estuary with the depths of ≤19 m. Thus, it is plausible to select the most dominant Group A (high ammonia cluster, HAC) OTU sequence (KY387998) to construct standard curves for qPCR measurements in the estuary.

2. Thus, the authors should mention about the possibility that the interpretation of the niche separation among AOA subgroups may be influenced by PCR bias in this manuscript.

Response: Agree. We added the statement in the paragraph of AOA clone library analysis in 3.3 section. - "Although the niche separation among AOA subgroups may be influenced by some bias during PCR amplification, overall distribution of HAC and LAC subgroups are plausible."

(HAC, high ammonia cluster; LAC, low ammonia cluster)

3. The authors obtained archaeal amoA gene sequences in the clone analysis and then used the selected sequences to obtain standard curves in the following qPCR. However, qPCR primer set for archaeal amoA genes was identical to a primer set used for the conventional clone analysis. This generally allows complete match between archaeal amoA clone sequences and primer sequences in PCR reaction for obtaining standard curves. In contrast, the presence of few mismatch residues is expected between environmental amoA gene sequences and primer sequences in qPCR. The gap would be a reason for underestimation of archaeal amoA genes in environmental samples. In qPCR of archaeal amoA, a primer set, Wuchter et al. 2006, or other primer set that does not overlap the annealing regions in the initial clone analysis is recommended.

Response: Primers are designed always based on existing sequences. So, there is always few mismatches between environmental sequences and primer sequences even

if we use different primer sets for qPCR and PCR, which would lead to underestimation of abundance. We admit it, but it is a normal issue in the field.

As for standards, selecting a most dominant OTU sequence in the studied region for standard curves construction should be the most reasonable selection.

Our qPCR measurements of archaeal amoA gene were only performed in the estuary (Peal River Estuary, PRE) in this manuscript. According to the reviewer's suggestion, we carried out qPCR using the pairs of primers Arch-amoA-for/Arch-amoA-rev (WuchterAOA, targeting 'high-ammonia concentration' archaeal amoA, HAC) from Wuchter et al. (2006) for all samples of the PRE, and Arch-amoAFA/Arch-amoAR (BemanAOA, targeting 'shallow' clades, group A) from Beman et al. (2008) for six samples at sites P6 (upper reaches), P8 (middle reaches), P11 and P12 (lower reaches) of the PRE.

(1) The results show that the abundances of archaeal amoA gene measured by BemanAOA primer set were similar to those measured by FranAOA primer set used in our manuscript (Fig. R1).

(2) However, WuchterAOA primer set cannot detect archaeal amoA gene for the samples from hypoxic zone (the upper reaches of the PRE), obtained similar abundance in the middle reaches of the PRE, and retrieved higher abundance of archaeal amoA gene from the samples in the lower reaches of the PRE (Table R1). Thus, we replaced the data (using FranAOA primer set) at the lower reaches sites P9-12 with the new abundance data (using WuchterAOA primer set) of archaeal amoA gene in the revised manuscript. We also revised the related statements. Overall, the conclusions based on the new data set are consistent with the previous ones. Standard curves were shown in Fig. R2.

P2, L3: We analyzed diversity and abundance of ammonia-oxidizing archaea (AOA) and betaproteobacteria (AOB), nitrite-oxidizing bacteria (NOB), and nitrification rates to

Response: Revised as suggested.

P2, L6-7: AOA were generally more abundant than betaproteobacterial AOB, however,

Response: Revised as suggested.

P2, L12: What does "a coupling of ammonia and nitrite oxidizers" mean?

Response: Sorry for the unclear sentence. We revised as "There is a significant positive correlation between ammonia and nitrite oxidizer abundances in the hypoxic waters of the estuary, suggesting a possible coupling through metabolic interactions between them."

P4, L2-14: Please insert a sentence to present the close relationship between Nitrospina and "Ca. Nitromaritima". Sequences belong to "Ca. Nitromaritima" had been reported as a group in Nitrospina until the definition of "Ca. Nitromaritima". Thus, the discussion in Lucker et al. 2013 includes both Nitrospina and "Ca. Nitromaritima".

Response: Thanks for the reviewer's suggestion. We supplied this information in the revised version. - "Candidatus Nitromaritima were recently identified based on metagenomic data in Red Sea brines (Ngugi et al., 2016), which were previously reported as a group in Nitrospina."

P4, L6: Information from Pachiadaki et al. 2017 should be referred through the manuscript.

Response: We added the citation of "Pachiadaki et al. 2017" in the revised manuscript.

P6, L5: Strains and/or genomic DNA from the public repositories used in this study should be summarized as the first paragraph in Materials and methods.

Response: Thanks for the reviewer's suggestion. We added this paragraph in Materials and methods. Please see the below.

2.1 Strains and genomic DNAs

We obtained strains Cadidatus Nitrospira defluvii A17 and Nitrospina gracilis 3/211 and their genomic DNAs from the University of Hamburg, Germany. The full-length 16S rRNA gene fragments were used as the standards for construction of standard curves during qPCR amplification.

P8, L18-: Did the authors determine OTUs for each library or among the libraries obtained in this study ?

Response: We revised this sentence as "all sequences among the libraries for each gene were grouped into operational taxonomic units (OTUs) based on a 5% sequence divergence cutoff ."

P8, L18-: Did the authors conduct any chimera check programs? It has been reported that more than 10% of the archaeal amoA gene sequences in the public database are chimera sequences (Eloy Alves et al. 2018).

Response: Thanks for the reviewer's suggestion. We did chimera check through Bellerophon and manual BLASTp analysis. We added this statement in Materials and methods. Please see the below.

"All sequences were analyzed with Bellerophon program (http://comp-bio.anu.edu.au/bellerophon/bellerophon.pl) to detect chimeric sequences in multiple sequences alignments (Huber et al., 2004). The putative chimeras were further checked manually through BLASTp analysis to verify whether these were chimeras."

P9, L13: Names of the sequence used to obtain standard curves for the qPCR should be presented.

Response: We added the accession numbers for the sequences used to obtain standard curves. -"Standard curves were constructed for archaeal and $\beta$-proteobacterial amoA genes using plasmid DNA (accession numbers KY387998 for AOA and MH458281 for AOB) from clone libraries."

P9, L14: How did the authors obtain DNA fragments?

Response: Genomic DNAs of Cadidatus Nitrospira defluvii A17 and Nitrospina gracilis 3/211 were obtained from Professor Eva Spieck from the University of Hamburg, Germany. We added a paragraph on strains and genomic DNAs in Materials and methods. Please see the below.

2.1 Strains and genomic DNAs

We obtained strains Cadidatus Nitrospira defluvii A17 and Nitrospina gracilis 3/211 and their genomic DNAs from the University of Hamburg, Germany. The full-length 16S rRNA gene fragments were used as the standards for construction of standard curves during qPCR amplification.

P11, L5: Pseudomonas chlororaphis subsp. aureofaciens (ATCC 13985)

Response: Thanks. Revised as suggested.

P13, L3-: Did the authors obtain data of turbidity or light intensity during the sampling?

Response: We did not obtain data of turbidity and light intensity. But we showed the data of total suspended material (TSM) concentrations, which can reflect the turbidity and light intensity.

P15, L11-: Please clarify how many clone libraries constructed for each gene. Supplementary tables presenting distribution of OTUs will help readers to understand the results.

Response: We described clone libraries constructed for each gene in 3.2 section. -"Archaeal and $\beta$-proteobacterial amoA and NOB (Nitrospira, Nitrospina, and Nitrobacter) nxrB gene clone libraries were constructed for the FL communities from the surface and bottom waters at site P8 and P9 because the most dramatic variations in biogeochemical properties along the PRE transect were present between these two sites (Fig. 2). In addition, archaeal amoA gene clone libraries were constructed at 75, 200, 800, and 3000 m water depth from SEATS, while a NOB Nitrospina nxrB gene clone library was constructed only at 800 m at SEATS as genes were not amplified successfully at

the other three water depths."

According to the reviewer's suggestion, we added the number of the clone libraries for each gene in Table S4.

P18, L10: Please present the values of detection limits in each qPCR if possible.

Response: Supplied these values in Table S3 as suggested.

P19, L6: As I know, the abundance of ammonia oxidizers is generally higher than nitrite oxidizers in aquatic environments. I am afraid that the result was influenced by the technical issues described above.

Response: The abundance of ammonia oxidizers is generally higher than nitrite oxidizers in the oxygenated oceanic water column. However, in oxygen-deficient waters, NOB can reach high abundances exceeding ammonia oxidizers. For example, Füssel et al. (2012) and Beman et al. (2013) observed highly abundant Nitrospina and Nitrococcus in oceanic OMZs. We discussed this content in 4.2 section (Coupling between ammonia and nitrite oxidizers in the estuarine hypoxic niche).

For the technical issues, please see the response above. We also verified qPCR results using additional two published primer sets. Please see Table R1 and Figure R1 above.

P21, L5: dominant NOB.

Response: Revised as suggested.

P21, L5-: Information from Hawley et al. 2014 should be integrated in this discussion.

Response: We supplied the information from Hawley et al. 2014 in the revised manuscript. -"With metaproteomic analysis, Hawley et al. (2014) reported higher expression of NXR from NOB Nitrospira and Nitrospina than that of Amo from Thaumarchaeota in an oxygen-deficient water column, Saanich Inlet, British Columbia."

P25, L17: "availability of ammonia" or "ammonia concentration/flux" would be better

than "ammonia levels".

Response: We revised "ammonia levels" as " ammonia concentration/flux (Sintes et al., 2013; 2016; Nunoura et al., 2015)".

P25, L17: Appropriate references should be provided.

Response: Added.

P26, L12: Appropriate references for light inhibition on the growth of nitrifiers should be provided.

Response: We added two citations. Please see below.

Lomas, M. W., and Lipschultz, F.: Forming the primary nitrite maximum: Nitrifiers or phytoplankton?, Limnol. Oceanogr., 51, 2453–2467, 2006.

Merbt, S. N., Stahl, D. A., Casamayor, E. O., Martí, E., Nicol, G. W., and Prosser, J. I.: Differential photoinhibition of bacterial and archaeal ammonia oxidation, FEMS Microbiol. Lett., 327, 41–46, 2012.

References: Proc. Natl. Acad. Sci. USA instead of P. Natl. Acad. Sci. USA

Response: Revised throughout the references list.

Fig. 5: Did the Nitromaritima sequence excluded in this phylogenetic analysis?

Response: Thanks for the reviewer's suggestion. We reconstructed the phylogenetic tree of Nitrospina, in which two nitrite oxidoreductase beta subunits (nxrB) gene sequences of Candidatus Nitromaritima were included. Please see the Figure 5 in the revised manuscript.

Please refer to the attached Supplements for Table R1.

Please also note the supplement to this comment:
https://www.biogeosciences-discuss.net/bg-2018-189/bg-2018-189-AC1-

supplement.pdf

[Figure]

[Figure]

**Fig. 1.** Figure R1. The archaeal amoA gene copies from six samples at sites P6 (upper reaches), P8 (middle reaches), P11 and P12 (lower reaches) of PRE using the two primer sets. *, No DNAs.

[Figure]

**Fig. 2.** Figure R2. Standard curves for the BemanAOA and WuchterAOA primer sets qPCR measurements.

**Supplement:**

**Table R1. Archaeal *amo*A gene copies using FranAOA primer set versus WuchterAOA primer set.**

| Station | Water Depth (m) | Sampling Depth (m) | Archaeal *amo*A (copies L$^{-1}$) FranAOA | | | | Archaeal *amo*A (copies L$^{-1}$) Wuchter AOA | | | |
|---|---|---|---|---|---|---|---|---|---|---|
| | | | FL | SD | PA | SD | FL | SD | PA | SD |
| P1 | 8.9 | 1 | 0 | | $1.50\times10^3$ | $4.00\times10$ | ND | | ND | |
| | | 7 | 0 | | $1.25\times10^3$ | $5.72\times10$ | ND | | ND | |
| P2 | 9.8 | 1 | NS | | | | NS | | | |
| | | 7 | $2.77\times10^3$ | $2.69\times10$ | $4.46\times10^3$ | $1.24\times10^3$ | ND | | ND | |
| P3 | 10.2 | 1 | NS | | | | NS | | | |
| | | 8 | $2.56\times10^3$ | $2.51\times10^2$ | $1.13\times10^4$ | $8.45\times10$ | ND | | ND | |
| P4 | 21.5 | 1 | NS | | | | NS | | | |
| | | 18 | $6.57\times10^2$ | $2.18\times10$ | $1.21\times10^4$ | $5.16\times10^2$ | ND | | ND | |
| P5 | 22.5 | 1 | $4.10\times10^3$ | $8.00\times10$ | $6.54\times10^3$ | $3.00\times10$ | ND | | ND | |
| | | 19 | $3.26\times10^3$ | $1.09\times10^2$ | $9.16\times10^3$ | $3.27\times10^2$ | ND | | ND | |
| P6 | 18.8 | 1 | $3.62\times10^3$ | $3.05\times10^2$ | $1.12\times10^4$ | $2.41\times10^2$ | ND | | ND | |
| | | 16 | NS | | | | NS | | | |
| P7 | 12 | 1 | $4.07\times10^4$ | $2.18\times10^3$ | $1.09\times10^5$ | $4.43\times10^3$ | ND | | ND | |
| | | 10 | $1.02\times10^4$ | $2.23\times10^3$ | $8.27\times10^3$ | $6.77\times10^2$ | $1.25\times10^4$ | $3.69\times10^2$ | $1.41\times10^4$ | $1.67\times10^3$ |
| P8 | 5 | 1 | $2.61\times10^3$ | $4.72\times10$ | $6.44\times10^4$ | $3.10\times10^3$ | ND | | $6.86\times10^4$ | $8.14\times10^3$ |
| | | 3.5 | $2.90\times10^3$ | $2.72\times10^2$ | $4.95\times10^4$ | $4.52\times10^3$ | ND | | ND | |
| P9 | 8 | 1 | $1.02\times10^3$ | $5.51\times10$ | $2.39\times10^4$ | $1.72\times10^3$ | $2.04\times10^4$ | $1.10\times10^3$ | $3.54\times10^4$ | $2.54\times10^3$ |
| | | 6 | $5.01\times10^2$ | $2.19\times10$ | $4.54\times10^5$ | $1.67\times10^4$ | $1.01\times10^4$ | $4.42\times10^2$ | $6.82\times10^5$ | $2.51\times10^4$ |
| P10 | 12.9 | 1 | $1.11\times10^3$ | $2.75\times10$ | $7.30\times10^2$ | $2.31\times10^2$ | $7.20\times10^4$ | $2.10\times10^4$ | $2.55\times10^4$ | $4.63\times10^3$ |
| | | 11 | $3.68\times10^3$ | $1.60\times10^2$ | $9.48\times10^3$ | $1.63\times10^3$ | $1.13\times10^5$ | $4.92\times10^3$ | $1.86\times10^5$ | $3.20\times10^4$ |
| P11 | 14.2 | 1 | 0 | | 0 | | $1.44\times10^4$ | $2.52\times10^3$ | $7.75\times10^2$ | $8.12\times10$ |
| | | 12 | $4.71\times10^3$ | $4.35\times10^2$ | $1.83\times10^4$ | $1.14\times10^3$ | $1.30\times10^5$ | $6.30\times10^3$ | $1.37\times10^5$ | $1.48\times10^4$ |
| P12 | 16 | 1 | $7.14\times10^2$ | $5.22\times10$ | 0 | | $1.21\times10^4$ | $1.92\times10^3$ | $4.72\times10^3$ | $9.84\times10^2$ |
| | | 14 | $2.18\times10^4$ | $1.73\times10^3$ | $3.30\times10^4$ | $3.98\times10^3$ | $3.02\times10^5$ | $7.81\times10^4$ | $2.41\times10^5$ | $4.90\times10^3$ |

FL, free-living; PA, particle-associated; NS, no sample; ND, not detected (We tried various optimization strategies for qPCR.)

---

## Referee Comment (RC2) · Anonymous Referee #2 · 28 Jun 2018

Hou et al. investigated the distribution of Ammonia and Nitrite oxidizers in a subtropical estuary of China by using the functional gene-based clone library and qPCR analyses as well as the determination of nitrification rates. The main conclusion of this work is that substrate affinity/preference of Ammonia and Nitrite oxidizers may play an important role in determining their distribution patterns in estuarine-ocean gradient. Some small comments are provided for improving this manuscript.

1. page 1, Line 2, and page 1, Line 15, I think "between" should be changed into "of"; 2. page 1, Line 15-18, Please add some background information related to niche differentiation of ammonia and nitrite oxidizers. This may facilitate readers to get a

quick view of the current research status. 3. page 8, Line 9, could you provide a coverage information about this primer set you designed? 4. page 13, Line 6, reference citations? or based on your results? 5. page 16, Line 5, I think the group E of AOA belong to the typical Soil/Sediment cluster, while other groups you defined belong to the typical Water/Sediment cluster. Actually, the HAC and LAC clusters were defined on the basis of the later one, especially for the Marine cluster within the Water/Sediment cluster. If you want to define the members within Soil/Sediment cluster, like group E, please provide more supporting evidence/cited references. 6.

---

## Author Comment (AC2) · 20 Jul 2018

Because of inserted and formatted figures and table, you will find all the responses in the supplement file. Thank you for your understanding. Y. Zhang and co-authors

Please also note the supplement to this comment:
https://www.biogeosciences-discuss.net/bg-2018-189/bg-2018-189-AC2-supplement.pdf

---

## Author Comment (AC4) · 20 Jul 2018

**Manuscript Number: bg-2018-189**

**Manuscript title: Niche differentiation of ammonia and nitrite oxidizers along a salinity gradient from the Pearl River estuary to the South China Sea**

**Response to Reviewer #2**

**We greatly thank the reviewer for the valuable comments, useful suggestions and careful revisions, based on which we have revised the manuscript. And the point-by-point responses to the comments are in blue colour as follows.**

**Anonymous Referee #2**

Hou et al. investigated the distribution of Ammonia and Nitrite oxidizers in a subtropical estuary of China by using the functional gene-based clone library and qPCR analyses as well as the determination of nitrification rates. The main conclusion of this work is that substrate affinity/preference of Ammonia and Nitrite oxidizers may play an important role in determining their distribution patterns in estuarine-ocean gradient. Some small comments are provided for improving this manuscript.

1. page 2, Line 2, and page 4, Line 15, I think "between" should be changed into "of";

**Response:**

Revised as suggested.

2. page 4, Line 15-18, Please add some background information related to niche differentiation of ammonia and nitrite oxidizers. This may facilitate readers to get a quick view of the current research status.

**Response:**

Thanks for the reviewer's suggestion. We added some background information related to niche differentiation of ammonia and nitrite oxidizers. — "*For example, both AOA and AOB are frequently found together in estuarine and coastal regimes and share the same ecosystem function (Bernhard et al., 2010; Zhang et al., 2014a), but in many situations, only AOA or AOB are predominant (Cébron et al., 2003; Hollibaugh et al., 2011; Li et al., 2014) as their physiological responses to environmental stressors may be different. Similarly, Nitrospira, Nitrospina, Nitrococcus, and/or Nitrobacter are frequently found together in estuarine and marine regimes, but there is no a consistent distribution pattern between them*

*(Cébron et al., 2005; Füssel et al., 2012; Nunoura et al., 2015; Pachiadaki et al., 2017), suggesting that niche partitioning and niche specialization support the coexistence of sympatric NOB. Moreover, between ammonia and nitrite oxidizers, there is a coupling in abundance and distribution in Monterey Bay and the North Pacific Subtropical Gyre (Mincer et al., 2007) or decoupling in Gulf of Mexico (Bristow et al., 2015).*"

3. page 8, Line 9, could you provide a coverage information about this primer set you designed?

**Response:**

Thanks for the reviewer's suggestion. There were only two *nxr*B gene sequences (from *Nitrospina gracilis* 3/211) in the NCBI database when the primer pair of nxrBNF and nxrBNR was designed, and the coverage is 100%. We discussed the coverage of this primer pair in Discussion 4.1 section and added the coverage analysis. — "*Among 23 sequences of Nitrospina nxrB genes available in the databases, only seven sequence could not be targeted by the primers nxrBNF and nxrBNR due to >3 mismatching bases for either primer, indicating a ~70% coverage of the primers (100% if allowing 5 mismatching bases).*"

4. page 13, Line 6, reference citations? or based on your results?

**Response:**

Sorry for the unclear sentence. The reference citation on the characteristics of the upper, middle and lower reaches of the PRE is Wang et al. (2012). We revised this sentence as "*The upper reaches receive a small amount of freshwater, sewage, and industrial effluent discharge; the middle reaches receive about half of the freshwater from the North and West rivers, tributaries of the Pearl River, with little salinity stratification; the lower reaches are controlled mainly by estuarine mixing of freshwater and seawater (Wang et al., 2012).*"

5. page 16, Line 5, I think the group E of AOA belong to the typical Soil/Sediment cluster, while other groups you defined belong to the typical Water/Sediment cluster. Actually, the HAC and LAC clusters were defined on the basis of the later one, especially for the Marine cluster within the Water/Sediment cluster. If you want to

define the members within Soil/Sediment cluster, like group E, please provide more supporting evidence/cited references.

**Response:**

Many thanks for the reviewer's suggestion. Indeed, group E belongs to Soil/sediment cluster. We added the related statement in 3.3 section — *"According to the framework of Francis et al. (2005), groups A, Ba, and Bb were defined as Water column cluster, group D was defined as Sediments cluster, and group E was defined as Soil/sediment cluster."* We also added the cluster information in Figure 3 and S3.

According to the framework of Sintes et al. (2013), there is a rough range of ammonia concentration for HAC (20 to 100 nM or even higher) and LAC (frequently below detection limit). Our field data on ammonia concentration confirmed the categorization of groups A (HAC), Ba and Bb (LAC), D (HAC), and E (HAC). We also added a reference citation to support that group E can be defined as HAC. — *"Tourna et al. (2011) and Hatzenpichler et al. (2008) have reported that two ammonia-oxidizing archaea Nitrososphaera viennensis and Nitrososphaera gargensis belonging to group E (crenarchaeal group I. 1b) tolerate high ammonia concentrations (1–15 mM and 0.14–3.08 mM, respectively)."*

---

## Author Response (AR1)

**Manuscript Number: bg-2018-189**

**Manuscript title: Niche differentiation of ammonia and nitrite oxidizers along a salinity gradient from the Pearl River estuary to the South China Sea**

**Response to Editor**

Comments to the Author:

Dear Dr. Hou

Your manuscript has now been seen by two referees. Although both reports are positive, one of the Referee # 1 has made some significant technical issues that need to be addressed by you. Although s/he has recommended minor revision, I would still send the revised manuscript to her/him for approval. Please take care of the comments. I look forward to receiving the revision at an early date.

Best regards

Wajih Naqvi

Dear Editor,

Thank you for taking the time to handle our manuscript and your assessment. We have carefully addressed each comment from two referees and tried our best to improve the manuscript according to their suggestions.

For some technical issues mentioned by Reviewer #1, please refer to our responses below for details. According to Reviewer #1's comment, we carried out qPCR using the pairs of primers Arch-amoA-for/Arch-amoA-rev (WuchterAOA) from Wuchter et al. (2006) for all samples of the PRE, and Arch-amoAFA/Arch-amoAR (BemanAOA) from Beman et al. (2008) for six samples at the upper reaches, middle reaches, lower reaches of the PRE. We replaced the data (using FranAOA primer set) at the lower reaches sites P9–12 with the new abundance data (using WuchterAOA primer set) of archaeal *amo*A gene in the revised manuscript, and also revised the related statements.

Our responses to all comments are listed below. We welcome any further comments. Thank you again for your time and kind efforts.

Best wishes

Yao Zhang

We greatly thank the reviewers for the valuable comments, useful suggestions and careful revisions, based on which we have revised the manuscript. And the point-by-point responses to the comments are in blue colour as follows.

**Response to Reviewer #1**

**T. NUNOURA**

**takuron@jamstec.go.jp**

The authors characterized the diversity, abundance and activity of nitrifiers associated with waters from the Pearl River estuary to the South China Sea. The data set provides novel insights into the niche separation and interactions of ammonia and nitrite oxidizers. However, I found technical issues in the quantification of archaeal ammonia oxidizers in the current version of manuscript as shown below.

In this study, archaeal *amo*A genes were used for a molecular marker for archaeal ammonia oxidizers, and there are technical issues in both diversity analysis and quantitative PCR. 1. The composition of archaeal *amo*A diversity is highly biased in PCR amplification comparing to the SSU rRNA gene analysis as reported previously (Meinhardt et al., 2015; Nunoura et al., 2015). In addition, it would lead inappropriate selection of sequences used to obtain standard curves in qPCR.

**Response:**

Meinhardt et al. (2015) showed that the archaeal *amo*A diversity retrieved from soil using their new-designed GenAOA primer set was more similar to that from the metagenomic data, compared with using FranAOA primer set, which was used in our manuscript. However, all the mainly different clades (*Nitrosotalea* 1.1, *Nitrososphaera* 1 (non-54d9), *Nitrososphaera* 1.1 and *Nitrososphaera* 54d9) between FranAOA clone library and metagenomic data were not adapted to the marine habitats.

Nunoura et al. (2015) showed that the distribution and abundance patterns of four subgroups of archaeal *amo*A genes (Group A, Ba, Bb, and D) in the Challenger Deep measured by qPCR were more similar to the MGI SSU rRNA gene clone library community structure, rather than the archaeal *amo*A gene clone library (using the FranAOA primer set). However, more specifically, above the Abyssal zone (4000–6000 m), the qPCR abundance patterns of the archaeal *amo*A subgroups were similar

to the archaeal *amo*A gene clone library analysis; the inconsistent pattern was present in the hadal zone (below 6000 m). Our clone library analysis of archaeal *amo*A gene was carried out at two estuary sites and in the South China Sea basin at 75 m, 200 m, 800 m, and 3000 m water depth. Our qPCR measurements of archaeal *amo*A gene were only performed in the estuary with the depths of ≤19 m. Thus, it is plausible to select the most dominant Group A (high ammonia cluster, HAC) OTU sequence (KY387998) to construct standard curves for qPCR measurements in the estuary.

2. Thus, the authors should mention about the possibility that the interpretation of the niche separation among AOA subgroups may be influenced by PCR bias in this manuscript.

**Response:**

Agree. We added the statement in the paragraph of AOA clone library analysis in 3.3 section. — "*Although the niche separation among AOA subgroups may be influenced by some bias during PCR amplification, overall distribution of HAC and LAC subgroups are plausible.*" (Page 18, Line 7–8)

(HAC, high ammonia cluster; LAC, low ammonia cluster)

3. The authors obtained archaeal *amo*A gene sequences in the clone analysis and then used the selected sequences to obtain standard curves in the following qPCR. However, qPCR primer set for archaeal *amo*A genes was identical to a primer set used for the conventional clone analysis. This generally allows complete match between archaeal *amo*A clone sequences and primer sequences in PCR reaction for obtaining standard curves. In contrast, the presence of few mismatch residues is expected between environmental *amo*A gene sequences and primer sequences in qPCR. The gap would be a reason for underestimation of archaeal *amo*A genes in environmental samples. In qPCR of archaeal *amo*A, a primer set, Wuchter et al. 2006, or other primer set that does not overlap the annealing regions in the initial clone analysis is recommended.

**Response:**

Primers are designed always based on existing sequences. So, there is always few mismatches between environmental sequences and primer sequences even if we use

different primer sets for qPCR and PCR, which would lead to underestimation of abundance. We admit it, but it is a normal issue in the field.

As for standards, selecting a most dominant OTU sequence in the studied region for standard curves construction should be the most reasonable selection.

Our qPCR measurements of archaeal *amo*A gene were only performed in the estuary (Peal River Estuary, PRE) in this manuscript. According to the reviewer's suggestion, we carried out qPCR using the pairs of primers Arch-amoA-for/Arch-amoA-rev (WuchterAOA, targeting 'high-ammonia concentration' archaeal *amo*A, HAC) from Wuchter et al. (2006) for all samples of the PRE, and Arch-amoAFA/Arch-amoAR (BemanAOA, targeting 'shallow' clades, group A) from Beman et al. (2008) for six samples at sites P6 (upper reaches), P8 (middle reaches), P11 and P12 (lower reaches) of the PRE.

(1) The results show that the abundances of archaeal *amo*A gene measured by BemanAOA primer set were similar to those measured by FranAOA primer set used in our manuscript (Fig. R1).

(2) However, WuchterAOA primer set cannot detect archaeal *amo*A gene for the samples from hypoxic zone (the upper reaches of the PRE), obtained similar abundance in the middle reaches of the PRE, and retrieved higher abundance of archaeal *amo*A gene from the samples in the lower reaches of the PRE (Table R1). Standard curves were shown in Fig. R2. Thus, we replaced the data (using FranAOA primer set) at the lower reaches sites P9–12 with the new abundance data (using WuchterAOA primer set) of archaeal *amo*A gene in the revised manuscript. We also revised the related statements. Overall, the conclusions based on the new data set are consistent with the previous ones.

Detailed revisions are listed below:

2.6 Quantitative PCR amplification:

 "*Standard curves were constructed for archaeal and β-proteobacterial amoA genes using plasmid DNA (accession numbers KY387998 (targeted by the primers Arch-amoAF and Arch-amoAR) and MH638327 (targeted by the primers Arch-amoA-for and Arch-amoA-rev) for AOA and MH458281 for AOB) from clone libraries.*" (Page 11, Line 1–3)

3.4 Abundance distribution of ammonia and nitrite oxidizers and nitrification rates:

"*Archaeal and β-proteobacterial amoA gene abundances varied from below detection limit to $6.82 \times 10^5$ copies $L^{-1}$ (PA community in the bottom water of site P9) and from below detection limit to $3.42 \times 10^4$ copies $L^{-1}$ (PA community in the bottom water of site P4), respectively.*" (Page 20, Line 5)

"*All of the genes were significantly more abundant in the PA than the FL communities (Wilcoxon, P <0.05–0.01) (Fig. 6e and f).*" (Page 20, Line 16, and Page 58)

"*The abundance of the NOB 16S rRNA genes rapidly decreased and the AOM amoA genes increased (Fig. 6g and h), and archaea and Nitrospina became the dominant ammonia and nitrite oxidizers, respectively (Fig. 6a–f).*" (Page 21, Line 10–12, and Page 58)

4.3 Succession of dominant nitrifier groups from the estuary to the open ocean:

Deleted "*In addition, the β-AOB amoA gene abundances were found to be significantly correlated to more environmental factors, including nitrite, nitrate, silicate, salinity, TSM, DO, and pH, in the PRE, whereas only one factor (TSM) was correlated to the AOA amoA gene (Table S5). We speculate that AOB could be better adapted to the estuarine habitat than AOA.*" (Page 25, Line 7)

4.4 Environmental parameters allowing niche differentiation:

Added "*Both nitrite and nitrate concentrations were negatively correlated to archaeal amoA gene abundance in the estuary (P <0.05–0.01, Table S5), which is consistent with the observations from the present study and previous studies that AOA are more dominant in oligotrophic environments (Wuchter et al., 2006; Newell et al., 2013).*" (Page 28, Line 5–8, and Supplement Page 9)

"*Notably, all genes were significantly positively correlated to TSM concentrations in PA communities (P <0.05–0.01, Table S5).*" (Page 28, Line 9–10, and Supplement Page 9)

Deleted "*This is consistent with the observation that all of the genes were significantly more abundant in the PA communities.*" (Page 28, Line 10)

Added "*pH was also negatively correlated to the β-AOB amoA and Nitrospira 16S rRNA gene abundances, but positively correlated to the archaeal amoA gene (P <0.05–0.01, Table S5). A similar observation was found by Li et al. (2011) in mangrove sediments at the northwestern corner of the New Territories of Hong Kong.*

*However, AOA and AOB amoA gene abundances were both previously found increasing with pH in soils (Gubry-Rangin et al., 2011) and the open ocean (Nunoura et al., 2015). This is probably related to lower availability of the substrate (ammonia) due to increased ionization to ammonium as pH decreases. In an estuary with sufficient nutrients, such as the PRE, negative correlations between gene abundances and pH could in fact be attributed to co-varying of pH with DO concentrations.*" (Page 29, Line 1–9, and Supplement Page 9)

"*In this study, silicate concentrations and salinity were found to be positively and negatively correlated, respectively, to the β-AOB amoA and Nitrospira 16S rRNA gene abundances; the opposite correlations were observed in archaeal amoA gene abundance (P <0.05–0.01, Table S5). These results suggest that β-AOB and Nitrospira recovered in the PRE could partly originate from the Pearl River or upstream and AOA could partly originate from the SCS.*" (Page 29, Line 14–17, and Supplement Page 9)

The revised figures and tables include Fig. 6, Fig. 9, Fig.11, Table S1, Table S3, and Table S5. Please see the revised manuscript and supplement. (Page 58, Page 62, Page 64, Supplement Page 1–2, Supplement Page 5 and Supplement Page 9)

**Table R1. Archaeal *amo*A gene copies using FranAOA primer set versus WuchterAOA primer set.**

| Station | Water Depth (m) | Sampling Depth (m) | Archaeal *amo*A (copies L$^{-1}$) FranAOA | | | | Archaeal *amo*A (copies L$^{-1}$) Wuchter AOA | | | |
|---|---|---|---|---|---|---|---|---|---|---|
| | | | FL | SD | PA | SD | FL | SD | PA | SD |
| P1 | 8.9 | 1 | 0 | | $1.50\times10^3$ | $4.00\times10$ | ND | | ND | |
| | | 7 | 0 | | $1.25\times10^3$ | $5.72\times10$ | ND | | ND | |
| P2 | 9.8 | 1 | NS | | | | NS | | | |
| | | 7 | $2.77\times10^3$ | $2.69\times10$ | $4.46\times10^3$ | $1.24\times10^3$ | ND | | ND | |
| P3 | 10.2 | 1 | NS | | | | NS | | | |
| | | 8 | $2.56\times10^3$ | $2.51\times10^2$ | $1.13\times10^4$ | $8.45\times10$ | ND | | ND | |
| P4 | 21.5 | 1 | NS | | | | NS | | | |
| | | 18 | $6.57\times10^2$ | $2.18\times10$ | $1.21\times10^4$ | $5.16\times10^2$ | ND | | ND | |
| P5 | 22.5 | 1 | $4.10\times10^3$ | $8.00\times10$ | $6.54\times10^3$ | $3.00\times10$ | ND | | ND | |
| | | 19 | $3.26\times10^3$ | $1.09\times10^2$ | $9.16\times10^3$ | $3.27\times10^2$ | ND | | ND | |
| P6 | 18.8 | 1 | $3.62\times10^3$ | $3.05\times10^2$ | $1.12\times10^4$ | $2.41\times10^2$ | ND | | ND | |
| | | 16 | NS | | | | NS | | | |
| P7 | 12 | 1 | $4.07\times10^4$ | $2.18\times10^3$ | $1.09\times10^5$ | $4.43\times10^3$ | ND | | ND | |
| | | 10 | $1.02\times10^4$ | $2.23\times10^3$ | $8.27\times10^3$ | $6.77\times10^2$ | $1.25\times10^4$ | $3.69\times10^2$ | $1.41\times10^4$ | $1.67\times10^3$ |
| P8 | 5 | 1 | $2.61\times10^3$ | $4.72\times10$ | $6.44\times10^4$ | $3.10\times10^3$ | ND | | $6.86\times10^4$ | $8.14\times10^3$ |
| | | 3.5 | $2.90\times10^3$ | $2.72\times10^2$ | $4.95\times10^4$ | $4.52\times10^3$ | ND | | ND | |
| P9 | 8 | 1 | $1.02\times10^3$ | $5.51\times10$ | $2.39\times10^4$ | $1.72\times10^3$ | $2.04\times10^4$ | $1.10\times10^3$ | $3.54\times10^4$ | $2.54\times10^3$ |
| | | 6 | $5.01\times10^2$ | $2.19\times10$ | $4.54\times10^5$ | $1.67\times10^4$ | $1.01\times10^4$ | $4.42\times10^2$ | $6.82\times10^5$ | $2.51\times10^4$ |
| P10 | 12.9 | 1 | $1.11\times10^3$ | $2.75\times10$ | $7.30\times10^2$ | $2.31\times10^2$ | $7.20\times10^4$ | $2.10\times10^4$ | $2.55\times10^4$ | $4.63\times10^3$ |
| | | 11 | $3.68\times10^3$ | $1.60\times10^2$ | $9.48\times10^3$ | $1.63\times10^3$ | $1.13\times10^5$ | $4.92\times10^3$ | $1.86\times10^5$ | $3.20\times10^4$ |
| P11 | 14.2 | 1 | 0 | | 0 | | $1.44\times10^4$ | $2.52\times10^3$ | $7.75\times10^2$ | $8.12\times10$ |
| | | 12 | $4.71\times10^3$ | $4.35\times10^2$ | $1.83\times10^4$ | $1.14\times10^3$ | $1.30\times10^5$ | $6.30\times10^3$ | $1.37\times10^5$ | $1.48\times10^4$ |
| P12 | 16 | 1 | $7.14\times10^2$ | $5.22\times10$ | 0 | | $1.21\times10^4$ | $1.92\times10^3$ | $4.72\times10^3$ | $9.84\times10^2$ |
| | | 14 | $2.18\times10^4$ | $1.73\times10^3$ | $3.30\times10^4$ | $3.98\times10^3$ | $3.02\times10^5$ | $7.81\times10^4$ | $2.41\times10^5$ | $4.90\times10^3$ |

FL, free-living; PA, particle-associated; NS, no sample; ND, not detected (We tried various optimization strategies for qPCR.)

[Figure]

**Figure R1.** The archaeal *amo*A gene copies from six samples at sites P6 (upper reaches), P8 (middle reaches), P11 and P12 (lower reaches) of PRE using the two primer sets. *, No DNAs.

[Figure]

**Figure R2.** Standard curves for the BemanAOA and WuchterAOA primer sets qPCR measurements.

P2, L3: We analyzed diversity and abundance of ammonia-oxidizing archaea (AOA) and betaproteobacteria (AOB), nitrite-oxidizing bacteria (NOB), and nitrification rates to

**Response:**

Revised as suggested. (Page 2, Line 3–4)

P2, L6-7: AOA were generally more abundant than betaproteobacterial AOB, however,

**Response:**

Revised as suggested. (Page 2, Line 6)

P2, L12: What does "a coupling of ammonia and nitrite oxidizers" mean?

**Response:**

Sorry for the unclear sentence. We revised as "*There is a significant positive correlation between ammonia and nitrite oxidizer abundances in the hypoxic waters of the estuary, suggesting a possible coupling through metabolic interactions between them.*" (Page 2, Line 11–13)

P4, L2-14: Please insert a sentence to present the close relationship between *Nitrospina* and "*Ca.* Nitromaritima". Sequences belong to "*Ca.* Nitromaritima" had been reported as a group in *Nitrospina* until the definition of "*Ca.* Nitromaritima". Thus, the discussion in Lucker et al. 2013 includes both *Nitrospina* and "*Ca.* Nitromaritima".

**Response:**

Thanks for the reviewer's suggestion. We supplied this information in the revised version. — "*Candidatus* Nitromaritima were recently identified based on metagenomic data in Red Sea brines (Ngugi et al., 2016), which were previously reported as a group in *Nitrospina*." (Page 4, Line 10–11)

P4, L6: Information from Pachiadaki et al. 2017 should be referred through the manuscript.

**Response:**

We added the citation of "*Pachiadaki et al. 2017*" in the revised manuscript. (Page 4, Line 6)

P6, L5: Strains and/or genomic DNA from the public repositories used in this study should be summarized as the first paragraph in Materials and methods.

**Response:**

Thanks for the reviewer's suggestion. We added this paragraph in Materials and methods. Please see the below.

*2.1 Strains and genomic DNAs*

*We obtained strains Candidatus Nitrospira defluvii A17 and Nitrospina gracilis 3/211 and their genomic DNAs from the University of Hamburg, Germany. The full-length 16S rRNA gene fragments were used as the standards for construction of standard curves during qPCR amplification.* (Page 6, Line 18 and Page 7, Line 1–3)

P8, L18-: Did the authors determine OTUs for each library or among the libraries obtained in this study ?

**Response:**

We revised this sentence as "*all sequences among the libraries for each gene were grouped into operational taxonomic units (OTUs) based on a 5% sequence divergence cutoff .*" (Page 10, Line 2–4)

P8, L18-: Did the authors conduct any chimera check programs? It has been reported that more than 10% of the archaeal *amo*A gene sequences in the public database are chimera sequences (Eloy Alves et al. 2018).

**Response:**

Thanks for the reviewer's suggestion. We did chimera check through Bellerophon and manual BLASTp analysis. We added this statement in Materials and methods. Please see the below.

"*All sequences were analyzed with Bellerophon program (http://comp-bio.anu.edu.au/bellerophon/bellerophon.pl) to detect chimeric sequences in multiple sequences alignments (Huber et al., 2004). The putative chimeras were further checked manually through BLASTp analysis to verify whether these were chimeras.*" (Page 9, Line 17–18 and Page 10, Line 1–2)

P9, L13: Names of the sequence used to obtain standard curves for the qPCR should be presented.

**Response:**

We added the accession numbers for the sequences used to obtain standard curves. — "*Standard curves were constructed for archaeal and β-proteobacterial amoA genes using plasmid DNA (accession numbers KY387998 (targeted by the primers Arch-amoAF and Arch-amoAR) and MH638327 (targeted by the primers Arch-amoA-for*

*and Arch-amoA-rev) for AOA and MH458281 for AOB) from clone libraries."* (Page 11, Line 1–3)

P9, L14: How did the authors obtain DNA fragments?

**Response:**

Genomic DNAs of Can*didatus* Nitrospira defluvii A17 and *Nitrospina gracilis* 3/211 were obtained from Professor Eva Spieck from the University of Hamburg, Germany. We added a paragraph on strains and genomic DNAs in Materials and methods. Please see the below.

*2.1 Strains and genomic DNAs*

*We obtained strains Candidatus Nitrospira defluvii A17 and Nitrospina gracilis 3/211 and their genomic DNAs from the University of Hamburg, Germany. The full-length 16S rRNA gene fragments were used as the standards for construction of standard curves during qPCR amplification.* (Page 6, Line 18 and Page 7, Line 1–3)

P11, L5: *Pseudomonas chlororaphis* subsp. *aureofaciens* (ATCC 13985)

**Response:**

Thanks. Revised as suggested. (Page 12, Line 12–13)

P13, L3-: Did the authors obtain data of turbidity or light intensity during the sampling?

**Response:**

We did not obtain data of turbidity and light intensity. But we showed the data of total suspended material (TSM) concentrations, which can reflect the turbidity and light intensity.

P15, L11-: Please clarify how many clone libraries constructed for each gene. Supplementary tables presenting distribution of OTUs will help readers to understand the results.

**Response:**

We described clone libraries constructed for each gene in 3.2 section. — *"Archaeal and β-proteobacterial amoA and NOB (Nitrospira, Nitrospina, and Nitrobacter) nxrB gene clone libraries were constructed for the FL communities from the surface and bottom waters at site P8 and P9 because the most dramatic variations in*

*biogeochemical properties along the PRE transect were present between these two sites (Fig. 2). In addition, archaeal amoA gene clone libraries were constructed at 75, 200, 800, and 3000 m water depth from SEATS, while a NOB Nitrospina nxrB gene clone library was constructed only at 800 m at SEATS as genes were not amplified successfully at the other three water depths.*"

According to the reviewer's suggestion, we added the number of the clone libraries for each gene in Table S4. (Supplement Page 8)

P18, L10: Please present the values of detection limits in each qPCR if possible.

**Response:**

The values of detection limits of all genes we measured were 2-3 copies $L^{-1}$. Supplied these values in Table S3 as suggested. (Supplement Page 5)

P19, L6: As I know, the abundance of ammonia oxidizers is generally higher than nitrite oxidizers in aquatic environments. I am afraid that the result was influenced by the technical issues described above.

**Response:**

The abundance of ammonia oxidizers is generally higher than nitrite oxidizers in the oxygenated oceanic water column. However, in oxygen-deficient waters, NOB can reach high abundances exceeding ammonia oxidizers. For example, Füssel et al. (2012) and Beman et al. (2013) observed highly abundant *Nitrospina* and *Nitrococcus* in oceanic OMZs. We discussed this content in 4.2 section (Coupling between ammonia and nitrite oxidizers in the estuarine hypoxic niche).

For the technical issues, please see the response above. We also verified qPCR results using additional two published primer sets. Please see Table R1 and Figure R1 above.

P21, L5: dominant NOB.

**Response:**

Revised as suggested. (Page 23, Line 4)

P21, L5-: Information from Hawley et al. 2014 should be integrated in this discussion.

**Response:**

We supplied the information from Hawley et al. 2014 in the revised manuscript. —

"*With metaproteomic analysis, Hawley et al. (2014) reported higher expression of*

*NXR from NOB Nitrospira and Nitrospina than that of Amo from Thaumarchaeota in an oxygen-deficient water column, Saanich Inlet, British Columbia.*" (Page 23, Line 5–7)

P25, L17: "availability of ammonia" or "ammonia concentration/flux" would be better than "ammonia levels".

**Response:**

We revised "*ammonia levels*" as " *ammonia concentration/flux*". (Page 27, Line 16)

P25, L17: Appropriate references should be provided.

**Response:**

Added. Please see below. (Page 27, Line 16)

Sintes, E., Bergauer, K., De Corte, D., Yokokawa, T., and Herndl, G. J.: Archaeal *amo*A gene diversity points to distinct biogeography of ammonia-oxidizing *Crenarchaeota* in the ocean, Environ. Microbiol., 15, 1647–1658, 2013. (Page 45, Line 4–6)

Sintes, E., De Corte, D., Haberleitner, E., and Herndl, G. J.: Geographic distribution of archaeal ammonia oxidizing ecotypes in the Atlantic Ocean, Front. Microbiol., 7, 1–14, 2016. (Page 45, Line 7–8)

Nunoura, T., Takaki, Y., Hirai, M., Shimamura, S., Makabe, A., Koide, O., Kikuchi, T., Miyazaki, J., Koba, K., Yoshida, N., Sunamura, M., and Takai, K.: Hadal biosphere: insight into the microbial ecosystem in the deepest ocean on Earth, Proc. Natl. Acad. Sci. USA, 112, E1230–E1236, 2015. (Page 42, Line 12–14)

P26, L12: Appropriate references for light inhibition on the growth of nitrifiers should be provided.

**Response:**

We added two citations. Please see below. (Page 28, Line 13–14)

Lomas, M. W., and Lipschultz, F.: Forming the primary nitrite maximum: Nitrifiers or phytoplankton?, Limnol. Oceanogr., 51, 2453–2467, 2006. (Page 40, Line 15–16)

Merbt, S. N., Stahl, D. A., Casamayor, E. O., Martí E., Nicol, G. W., and Prosser, J. I.: Differential photoinhibition of bacterial and archaeal ammonia oxidation, FEMS Microbiol. Lett., 327, 41–46, 2012. (Page 41, Line 11–13)

References: Proc. Natl. Acad. Sci. USA instead of P. Natl. Acad. Sci. USA

**Response:**

Revised throughout the references list. (Page 33, Line 17; Page 36, Line 6; Page 36, Line 16; Page 39, Line 9; Page 39, Line 17; Page 41, Line 1; Page 42, Line 14; Supplement Page 4, Line 4)

Fig. 5: Did the Nitromaritima sequence excluded in this phylogenetic analysis?

**Response:**

Thanks for the reviewer's suggestion. We reconstructed the phylogenetic tree of *Nitrospina*, in which two nitrite oxidoreductase beta subunits (*nxr*B) gene sequences of *Candidatus* Nitromaritima were included. Please see the revised Figure 5 (below). (Page 56)

[Figure]

**References:**

[revised manuscript text omitted]

**Comment [YZ5]:** Added as suggested. (RC1)

**Comment [YZ6]:** We supplied the information related to the close relationship between *Nitrospina* and *Ca.* Nitromaritima in the revised version. (RC1)

**Comment [YZ7]:** Revised as suggested. (RC2)

function (Bernhard et al., 2010; Zhang et al., 2014a), but in many situations, only AOA or AOB are predominant (Cébron et al., 2003; Hollibaugh et al., 2011; Li et al., 2014) as their physiological responses to environmental stressors may be different. Similarly, *Nitrospira*, *Nitrospina*, *Nitrococcus*, and/or *Nitrobacter* are frequently found together in estuarine and marine regimes, but there is no a consistent distribution pattern between them (Cébron et al., 2005; Füssel et al., 2012; Nunoura et al., 2015; Pachiadaki et al., 2017), suggesting that niche partitioning and niche specialization support the coexistence of sympatric NOB. Moreover, between ammonia and nitrite oxidizers, there is a coupling in abundance and distribution in Monterey Bay and the North Pacific Subtropical Gyre (Mincer et al., 2007) or decoupling in Gulf of Mexico (Bristow et al., 2015). A gradient from an estuary to the ocean, with various environmental gradients and distinct distribution patterns of various nutrient species, may provide diverse niches for the coexistence of microbial species (Martens-Habbena et al., 2009). It is thus an ideal system to study the niche differentiation of AOA, AOB and NOB and major controlling factors.

> **Comment [YZ8]:** We added some background information related to the niche differentiation of ammonia and nitrite oxidizers. (RC2)

[revised manuscript text omitted]

Comment [YZ15]: Group E belongs to Soil/sediment cluster. We added the related statement. (RC2)

Comment [YZ16]: We added a reference citation to support that group E can be defined as HAC. (RC2)

[revised manuscript text omitted]

**Comment [YZ18]:** We revised the maximum value of archaeal *amo*A gene abundance after replacing the data (using FranAOA primer set) at the lower reaches sites P9–12 with the new abundance data (using WuchterAOA primer set) of archaeal *amo*A gene in the revised manuscript. (RC1)

**Comment [YZ19]:** We reanalyzed the correlation after replacing the data (using FranAOA primer set) at the lower reaches sites P9–12 with the new abundance data (using WuchterAOA primer set) of archaeal *amo*A gene in the revised manuscript. (RC1)

1   below 2 mg L$^{-1}$ (Renaud, 1986), of the PRE upper reaches, have DO concentrations ranging from 0.19

2   to 1.93 mg L$^{-1}$ (Fig. 7). Generally, the abundance of NOB (sum of *Nitrospira* and *Nitrospina*) 16S rRNA

3   genes was significantly higher than the ammonia-oxidizing microbes (AOM, sum of archaea and

4   *β*-proteobacteria) *amo*A genes in the hypoxic waters (Wilcoxon, *P* <0.01; Fig. 6g and h). Notably,

5   significant positive relationships were observed between AOM and NOB groups for both the FL (Fig.

6   8a) and PA (Fig. 8b) communities (eight correlations, *P* <0.05–0.01, the findings were the same

7   excluding the maximum values), suggesting a coupling between ammonia and nitrite oxidizers in the

8   hypoxic estuarine niche.

9       The hypoxic zone gradually disappears seaward and the DO concentrations of sites P7–P12 varied

10   from 2.15 to 5.78 mg L$^{-1}$ (Fig. 7). The significant relationship between AOM and NOB collapsed

11   instantly. The abundance of the NOB 16S rRNA genes rapidly decreased and the AOM *amo*A genes

12   increased (Fig. 6g and h), and archaea and *Nitrospina* became the dominant ammonia and nitrite

13   oxidizers, respectively (Fig. 6a–f).

14       The nitrification rates generally decreased seaward with increasing DO concentrations, ranging

15   from 0.19 μmol L$^{-1}$ day$^{-1}$ in the bottom water (2 m above the seafloor) of site P9 to 75.81 μmol L$^{-1}$

16   day$^{-1}$ in the bottom water (3.5 m above the seafloor) of site P5 (Fig. 7). Distinctly higher nitrification

17   rates were observed in the hypoxic zone than the middle and lower reaches of the PRE (Wilcoxon

18   rank-sum test, *P* <0.05).
* * *
**Comment [YZ20]:** We revised the related statement after replacing the data (using FranAOA primer set) at the lower reaches sites P9–12 with the new abundance data (using WuchterAOA primer set) of archaeal *amo*A gene in the revised manuscript. (RC1)

**4 Discussion**

**4.1 Coverage of the primer pair for *Nitrospina nxr*B genes**

The primer pair of nxrBNF and nxrBNR targeting the *Nitrospina nxr*B genes was designed in this study according to two *nxr*B gene sequences of *N. gracilis* 3/211, which is the only isolated *Nitrospina* strain from the oxygenated ocean (Watson and Waterbury, 1971) and the only genome-sequenced *Nitrospina* so far (Lücker et al., 2013). Despite very few reference sequences, phylogenetic analysis of the *Nitrospina nxr*B gene sequences retrieved based on this primer pair indicated diverse phylogenetic taxa, including 12 OTUs and four major phylogenetic clusters. The relative abundances of the four groups showed that 77% of total sequences fell out of the 3/211 cluster (Fig. 5). Among 23 sequences of *Nitrospina nxr*B genes available in the databases, only seven sequences could not be targeted by the primers nxrBNF and nxrBNR due to >3 mismatching bases for either primer, indicating a ~70% coverage of the primers (100% if allowing 5 mismatching bases). Feng et al. (2016) and Rani et al. (2017) also designed primer pairs targeting *nxr*B and *nxr*A subunit genes of *Nitrospina*, respectively. However, Feng et al. (2016) did not obtain any *nxr*B target fragments and Rani et al. (2017) focused on the *nxr*A gene in marine sediments.

**Comment [YZ21]:** We discussed the coverage of the primer pair of nxrBNF and nxrBNR designed in this study. (RC2)

[revised manuscript text omitted]

**Comment [YZ38]:** Group E belongs to Soil/sediment cluster. We added the cluster information in Figure 3. (RC2)

[revised manuscript text omitted]

2 ᵃ, The primer set was used in the samples from the lower reaches (sites P9–12) of the estuary.

**Comment [YZ3]:** We supplied the values of detection limits in each gene qPCR. (RC1)

**Comment [YZ4]:** qPCR mixtures and conditions for WuchterAOA primer set. (RC1)

| Genes | No. of Libraries | n | No. of OTUs | C (%) | H' | 1/D | Chao1 |
|---|---|---|---|---|---|---|---|
| AOA *amo*A (SCS) | 4/4 | 392 | 60 | 0.94 | 3.04 | 10.64 | 49.46 |
| AOA *amo*A (PRE) | 4/4 | 127 | 23 | 0.90 | 2.13 | 4.87 | 42.5 |
| *β*-AOB *amo*A (PRE) | 2/4 | 26 | 3 | 0.96 | 0.43 | 1.28 | 3 |
| *Nitrospira nxr*B (PRE) | 4/4 | 345 | 29 | 0.96 | 1.79 | 3.57 | 42 |
| *Nitrospina nxr*B (PRE & SCS) | 4/8 | 185 | 12 | 0.98 | 1.79 | 4.83 | 12.75 |
| *Nitrobacter nxr*B (PRE) | 2/4 | 48 | 3 | 0.98 | 0.78 | 2.13 | 3 |

> **Comment [YZ5]:** We added the number of the clone libraries for each gene. (RC1)

3 n, number of sequences; OTU, operational taxonomic unit; C, coverage; H',

4 Shannon-Wiener Index; 1/D, Simpson's diversity Index; SCS, South China Sea; PRE,

5 Pearl River estuary. Numbers before slash indicate successful libraries; numbers after

6 slash indicate all amplified samples.

**Table S5.** *r* values for the relationship between gene abundances of nitrifiers and environmental parameters in the PRE.

| Gene | Community | Water mass parameters | | | Substrate parameters | | | Parameters influencing substrate availability | | |
|---|---|---|---|---|---|---|---|---|---|---|
| | | Temperature (n = 20) | Salinity (n = 20) | $SiO_3^{2-}$ (n = 20) | $NH_4^+$ (n = 15) | $NO_2^-$ (n = 20) | $NO_3^-$ (n = 20) | TSM (n = 19) | DO (n = 20) | pH (n = 20) |
| **AOB** *amo*A | **FL**[a] | 0.302 | −0.441 | 0.439 | −0.108 | 0.527* | 0.759** | −0.053 | −0.425 | −0.512* |
| | **PA**[b] | 0.332 | −0.474* | 0.475* | −0.048 | 0.706** | 0.464* | 0.520* | −0.525* | −0.496* |
| | **FL+PA** | 0.341 | −0.471* | 0.487* | −0.053 | 0.718** | 0.491* | 0.504* | −0.536* | −0.513* |
| **AOA** *amo*A | **FL**[a] | −0.754** | 0.691** | −0.709** | −0.376 | −0.461* | −0.728** | −0.203 | 0.412 | 0.585** |
| | **PA**[b] | −0.528* | 0.539* | −0.524* | −0.407 | −0.361 | −0.486* | 0.498* | 0.348 | 0.434 |
| | **FL+PA** | −0.717** | 0.703** | −0.697** | −0.468 | −0.470* | −0.673** | 0.330 | 0.441 | 0.577** |
| *Nitrospira* **16S rRNA** | **FL**[a] | 0.426 | −0.580** | 0.537* | −0.205 | 0.643** | 0.772** | −0.099 | −0.464* | −0.625** |
| | **PA**[b] | 0.356 | −0.474* | 0.491* | −0.073 | 0.730** | 0.518* | 0.504* | −0.541* | −0.524* |
| | **FL+PA** | 0.367 | −0.475* | 0.503* | −0.080 | 0.743** | 0.539* | 0.493* | −0.550* | −0.540* |
| *Nitrospina* **16S rRNA** | **FL**[a] | 0.097 | −0.167 | 0.158 | −0.268 | 0.436 | 0.253 | −0.315 | −0.190 | −0.230 |
| | **PA**[b] | 0.108 | −0.134 | 0.162 | −0.105 | 0.453* | 0.173 | 0.822** | −0.276 | −0.221 |
| | **FL+PA** | 0.111 | −0.140 | 0.167 | −0.115 | 0.468* | 0.182 | 0.811** | −0.282 | −0.229 |

2 [a], Free-living; [b], Particle-associated; *, *P* < 0.05; **, *P* < 0.01; TSM, Total suspended material; DO, Dissolved oxygen

**Comment [YZ6]:** We revised the correlations between archaeal *amo*A gene abundance and various environmental factors after replacing the data (using FranAOA primer set) at the lower reaches sites P9–12 with the new abundance data (using WuchterAOA primer set) of archaeal *amo*A gene in the revised manuscript. (RC1)

[Figure]

**Figure S1.** Depth profiles of biogeochemical parameters at SEATS.

[Figure]

2 **Figure S2.** Rarefaction curves of (a) AOA and *β*-AOB *amo*A gene sequences and (b)

3 *Nitrospira*, *Nitrospina*, and *Nitrobacter nxr*B gene sequences. The curves were

4 generated at 95% DNA sequence identity.

[Figure]

1 **Figure S3.** Unrooted neighbor-joining (NJ) phylogenetic tree of the archaeal *amo*A

2 gene sequences (expanded view for group Ba and Bb (LAC)). Clone sequences from

3 this study are shown in bold and sequences sharing 95% DNA identity are grouped.

4 Phylogenetic relationships were bootstrapped 1000 times, and bootstrap values greater

5 than 50% are shown. The scale bar indicates 5% estimated sequence divergence.

**Comment [YZ7]:** Group E belongs to Soil/sediment cluster. We added the cluster information in Figure S3. (RC2)

---

## Author Response (AR2)

**Manuscript Number: bg-2018-189**

**Manuscript title: Niche differentiation of ammonia and nitrite oxidizers along a salinity gradient from the Pearl River estuary to the South China Sea**

**Response to Editor**

Comments to the Author:

Dear Dr. Hou

I am pleased to inform you that your manuscript will be accepted for publication after you have incorporated the minor changes suggested by the reviewer.

Best regards

Wajih Naqvi

Dear Editor,

Thank you again for taking the time to handle our manuscript. We have carefully revised the manuscript based on the comment from Referee #1. Our response to the comment is listed below.

Best wishes,

Yao Zhang

**Response to Reviewer #1**

**T. NUNOURA**

**takuron@jamstec.go.jp**

P17, L14-: The integration of *Nitrososphaera* cluster into group E is apparently inappropriate considering the phylogenetic topology in this figure.

**Response:**

We agree with the reviewer's comment. We divided Soil/sediment cluster into two clades based on the phylogenetic topology in the revised manuscript. The clade containing *Nitrososphaera* was defined as group E according to Nunoura et al. (2013); the other clade was defined as group F. Please refer to Figure 3 and S3. We also revised the related statements in the revised manuscript.

Detailed revisions are listed below:

3.3 Phylogenetic analysis of archaeal *amo*A and *Nitrospira* and *Nitrospina nxr*B genes

"*A total of 519 AOA amoA gene sequences were recovered and grouped into three clusters (six groups A, Ba, Bb, D, E, and F) based on phylogenetic analysis (Fig. 3 and S3). According to the framework of Francis et al. (2005), groups A, Ba, and Bb were defined as Water column cluster, group D was defined as Sediments cluster, and groups E and F were defined as Soil/sediment cluster.*" (Page 17, Line 1–4)

"*Another half of the sequences retrieved from the PRE fell into Soil/sediment cluster (groups E and F) and had an 86% to 100% DNA sequence identity with sequences recovered from high ammonia environments, such as soil, sediment, biofilters, rivers, lakes, and water treatment plants (Fig. 3).*" (Page 17, Line 11–14)

"*Thus, we defined groups E and F as HAC. The ammonium concentrations at sites where sequences were recovered further confirmed the categorization of groups A, Ba, Bb, D, E, and F. The sequences falling in groups A, D, E, and F (HAC) were retrieved from sites with ammonium concentrations of 0.032 to 8.09 μM with the exception of four sequences (group A) retrieved from 3000 m at SEATS (below detection limit).*" (Page 17, Line 17–18 and Page 18, Line 1–3)

4.4 Environmental parameters allowing niche differentiation

"*The CCA analysis based on clone libraries (Fig. 10a) further revealed that AOA HAC groups D, E, and F were under the constraint of high nutrient conditions and HAC group A was positively influenced by TSM to an extent.*" (Page 26, Line 15–17)

The revised figures include Fig. 3, Fig. 10a, and Fig. S3. Please see the revised manuscript and supplement. (Page 53, Page 64, and Supplement Page 12)

[revised manuscript text omitted]

**Comment [YZ3]:** According to the reviewer's comment, Soil/sediment cluster was divided into two groups (groups E and F). We revised the related statement. (RC1)

[revised manuscript text omitted]

Comment [YZ4]: According to the reviewer's comment, Soil/sediment cluster was divided into two groups (groups E and F). We revised the related statement. (RC1)

[revised manuscript text omitted]

**Comment [YZ5]:** According to the reviewer's suggestion, Soil/sediment cluster was divided into two groups (groups E and F). We update the categorization information in Figure 3. (RC1)

[revised manuscript text omitted]

[a], The primer set was used in the samples from the lower reaches (sites P9–12) of the estuary.

**Table S4.** Diversity indices of AOA and *β*-AOB *amo*A, *Nitrospira*, *Nitrospina*, and

*Nitrobacter nxr*B genes based on 5% nucleic acid sequences cutoff.

| Genes | No. of Libraries | n | No. of OTUs | C (%) | H' | 1/D | Chao1 |
|---|---|---|---|---|---|---|---|
| AOA *amo*A (SCS) | 4/4 | 392 | 60 | 0.94 | 3.04 | 10.64 | 49.46 |
| AOA *amo*A (PRE) | 4/4 | 127 | 23 | 0.90 | 2.13 | 4.87 | 42.5 |
| *β*-AOB *amo*A (PRE) | 2/4 | 26 | 3 | 0.96 | 0.43 | 1.28 | 3 |
| *Nitrospira nxr*B (PRE) | 4/4 | 345 | 29 | 0.96 | 1.79 | 3.57 | 42 |
| *Nitrospina nxr*B (PRE & SCS) | 4/8 | 185 | 12 | 0.98 | 1.79 | 4.83 | 12.75 |
| *Nitrobacter nxr*B (PRE) | 2/4 | 48 | 3 | 0.98 | 0.78 | 2.13 | 3 |

n, number of sequences; OTU, operational taxonomic unit; C, coverage; H',

Shannon-Wiener Index; 1/D, Simpson's diversity Index; SCS, South China Sea; PRE,

Pearl River estuary. Numbers before slash indicate successful libraries; numbers after slash indicate all amplified samples.

Table S5. *r* values for the relationship between gene abundances of nitrifiers and environmental parameters in the PRE.

| Gene | Community | Water mass parameters | | | Substrate parameters | | | Parameters influencing substrate availability | | |
|---|---|---|---|---|---|---|---|---|---|---|
| | | Temperature (n = 20) | Salinity (n = 20) | $SiO_3^{2-}$ (n = 20) | $NH_4^+$ (n = 15) | $NO_2^-$ (n = 20) | $NO_3^-$ (n = 20) | TSM (n = 19) | DO (n = 20) | pH (n = 20) |
| AOB *amo*A | FL[a] | 0.302 | −0.441 | 0.439 | −0.108 | 0.527* | 0.759** | −0.053 | −0.425 | −0.512* |
| | PA[b] | 0.332 | −0.474* | 0.475* | −0.048 | 0.706** | 0.464* | 0.520* | −0.525* | −0.496* |
| | FL+PA | 0.341 | −0.471* | 0.487* | −0.053 | 0.718** | 0.491* | 0.504* | −0.536* | −0.513* |
| AOA *amo*A | FL[a] | −0.754** | 0.691** | −0.709** | −0.376 | −0.461* | −0.728** | −0.203 | 0.412 | 0.585** |
| | PA[b] | −0.528* | 0.539* | −0.524* | −0.407 | −0.361 | −0.486* | 0.498* | 0.348 | 0.434 |
| | FL+PA | −0.717** | 0.703** | −0.697** | −0.468 | −0.470* | −0.673** | 0.330 | 0.441 | 0.577** |
| *Nitrospira* 16S rRNA | FL[a] | 0.426 | −0.580** | 0.537* | −0.205 | 0.643** | 0.772** | −0.099 | −0.464* | −0.625** |
| | PA[b] | 0.356 | −0.474* | 0.491* | −0.073 | 0.730** | 0.518* | 0.504* | −0.541* | −0.524* |
| | FL+PA | 0.367 | −0.475* | 0.503* | −0.080 | 0.743** | 0.539* | 0.493* | −0.550* | −0.540* |
| *Nitrospina* 16S rRNA | FL[a] | 0.097 | −0.167 | 0.158 | −0.268 | 0.436 | 0.253 | −0.315 | −0.190 | −0.230 |
| | PA[b] | 0.108 | −0.134 | 0.162 | −0.105 | 0.453* | 0.173 | 0.822** | −0.276 | −0.221 |
| | FL+PA | 0.111 | −0.140 | 0.167 | −0.115 | 0.468* | 0.182 | 0.811** | −0.282 | −0.229 |

[a], Free-living; [b], Particle-associated; *, *P* < 0.05; **, *P* < 0.01; TSM, Total suspended material; DO, Dissolved oxygen

[Figure]

**Figure S1.** Depth profiles of biogeochemical parameters at SEATS.

[Figure]

**Figure S2.** Rarefaction curves of (a) AOA and *β*-AOB *amo*A gene sequences and (b)

*Nitrospira*, *Nitrospina*, and *Nitrobacter nxr*B gene sequences. The curves were generated at 95% DNA sequence identity.

[Figure]

**Figure S3.** Unrooted neighbor-joining (NJ) phylogenetic tree of the archaeal *amo*A

gene sequences (expanded view for group Ba and Bb (LAC)). Clone sequences from this study are shown in bold and sequences sharing 95% DNA identity are grouped.

Phylogenetic relationships were bootstrapped 1000 times, and bootstrap values greater than 50% are shown. The scale bar indicates 5% estimated sequence divergence.

**Comment [YZ1]:** According to the reviewer's suggestion, Soil/sediment cluster was divided into two groups (groups E and F). We update the categorization information in Figure S3. (RC1)